# Understanding MARS: When Scaling Momentum Correction Provably Helps

**Egor Shulgin** [1]  **Tamaz Gadaev** [2]  **Sarit Khirirat** [1]  **Peter Richtárik** [1]

## Abstract

MARS (Yuan et al., 2025) has recently emerged as a strong optimizer for large language model (LLM) training by scaling the correction term in momentum-based variance reduction (MVR). However, existing theory does not explain why this modification can improve convergence over the unscaled MVR choice $\gamma = 1$. In this paper, we provide a theoretical explanation for this phenomenon. We introduce $\gamma$-*similarity*, a refined similarity condition that captures how the scaling coefficient interacts with the stochastic gradient-difference structure. This condition recovers standard similarity at $\gamma = 1$ and smoothness at $\gamma = 0$. Using $\gamma$-similarity, we derive convergence guarantees for fixed-$\gamma$ MARS whose complexity depends explicitly on $\gamma$ and the corresponding $\gamma$-similarity constant. The bound reveals why small values of $\gamma$ can be beneficial: they may reduce the similarity term enough to outweigh the penalty from deviating from MVR. We prove that optimizing $\gamma$ gives MARS a lower complexity guarantee than MVR. Experiments with MARS-AdamW on GPT-style LLM pretraining corroborate the theory, showing that properly chosen small values of $\gamma$ improve token efficiency over $\gamma = 1$ and AdamW.

## 1. Introduction

The success of deep learning has driven significant attention toward nonconvex stochastic optimization problems of the form:

$$\min_{x \in \mathbb{R}^d} f(x) := \mathbb{E}_\xi[f_\xi(x)], \tag{1}$$

where $f_\xi(x)$ is a possibly nonconvex function, $x \in \mathbb{R}^d$ represents high-dimensional parameters, and $\xi$ is a random variable from an unknown data distribution $\mathcal{D}$. This formulation is central to the training of deep neural network models. State-of-the-art models, such as GPT-5 (Bubeck et al., 2025), LLaMa-3 (Grattafiori et al., 2024), and DeepSeek-R1 (Guo et al., 2025), comprise billions of parameters, and are trained on massive datasets. For solving such huge-scale tasks, standard stochastic optimizers are Stochastic Gradient Descent (SGD), including its adaptive variants, such as Adam (Kingma, 2015) and AdamW (Loshchilov & Hutter, 2019). These algorithms are widely used, because they construct inexpensive gradient estimates using only a few data points at each iteration, thus making them both memory- and computationally efficient. Furthermore, to minimize smooth functions, SGD achieves an $\mathcal{O}(1/T^{1/4})$ convergence in the gradient norm (Ghadimi & Lan, 2013).

### 1.1. Variance Reduction

To improve the convergence of SGD, various techniques have been proposed. One variance-reduction technique form an estimator combining stochastic gradients with periodically computed full gradients. Popular stochastic algorithms using this technique are Stochastic Variance Reduced Gradient (SVRG) algorithms (Johnson & Zhang, 2013; Konečný & Richtárik, 2013; Reddi et al., 2016). Despite their theoretical advantages over SGD, SVRG algorithms have achieved limited empirical success in solving huge-scale, nonconvex neural network training tasks. However, in these tasks, computing full gradients is infeasible, and the algorithms using this gradient estimator fail to effectively reduce gradient variance, as shown by Defazio & Bottou (2019). To improve the empirical training efficiency of SVRG, Yin et al. (2025) recently introduced a multiplicative coefficient into the variance-reduced gradient estimator.

Another prominent variance-reduction technique that does not require full gradients is Polyak momentum. A provably variance-reduced momentum variant is known as Momentum-based Variance Reduction (MVR) (Cutkosky & Orabona, 2019). MVR incorporates stochastic gradient difference as the correction term into the momentum update.

[1]King Abdullah University of Science and Technology (KAUST), Thuwal, Saudi Arabia [2]Independent Researcher. Correspondence to: Egor Shulgin <shulgin.yegor@gmail.com>.

*Proceedings of the 43rd International Conference on Machine Learning*, Seoul, South Korea. PMLR 306, 2026. Copyright 2026 by the author(s).

Specifically, it performs the iterations

$$x_{t+1} = x_t - \eta g_t, \quad \text{where} \tag{2}$$
$$g_t = (1 - \beta)(g_{t-1} + [\nabla f_{\xi_t}(x_t) - \nabla f_{\xi_t}(x_{t-1})])$$
$$+ \beta \nabla f_{\xi_t}(x_t). \tag{3}$$

Here, $\eta \geq 0$ is the stepsize, and $\beta \in [0, 1]$ is the momentum parameter. Note that MVR reduces to SGD when $\beta = 1$, and SGD with momentum when we omit the correction term $\nabla f_{\xi_t}(x_t) - \nabla f_{\xi_t}(x_{t-1})$. Furthermore, MVR achieves the $\mathcal{O}(1/T^{1/3})$ convergence in the gradient norm, which improves upon the $\mathcal{O}(1/T^{1/4})$ rate of SGD and SGD with momentum. Its convergence guarantee nearly matches the lower bounds originally established by Arjevani et al. (2023) and recently tightened by Fradin et al. (2026).

### 1.2. MARS

Motivated by recent advances in the use of the multiplicative coefficient in SVRG (Yin et al., 2025), Yuan et al. (2025) proposed the Momentum with Adaptive Residual Scaling (MARS) algorithm, which further enhances MVR by explicitly scaling the momentum correction term. Specifically, MARS uses a coefficient $\gamma \geq 0$ to control the strength of the momentum correction term in MVR, thus resulting in the update

$$g_t = (1 - \beta)(g_{t-1} + \gamma \Delta_t) + \beta \nabla f_{\xi_t}(x_t), \tag{4}$$

where $\Delta_t = \nabla f_{\xi_t}(x_t) - \nabla f_{\xi_t}(x_{t-1})$. A full description of MARS is provided in Algorithm 1. Moreover, MARS encompasses both MVR and SGD with momentum. In particular, MARS reduces to MVR when $\gamma = 1$, and to SGD with momentum when $\gamma = 0$.

---

**Algorithm 1** MARS ($\gamma$-MVR)

---

1: **Input:** stepsize $\eta > 0$; correction scale $\gamma \geq 0$; momentum $\beta \in (0, 1]$; initial point and gradient estimator $x_0, g_0 \in \mathbb{R}^d$.
2: **for** $t = 0, 1, \ldots, T - 1$ **do**
3:     Compute $x_{t+1} = x_t - \eta g_t$.
4:     Sample $\xi_{t+1}$.
5:     Compute $\Delta_{t+1} = \nabla f_{\xi_{t+1}}(x_{t+1}) - \nabla f_{\xi_{t+1}}(x_t)$.
6:     Update

$$g_{t+1} = (1 - \beta)(g_t + \gamma \Delta_{t+1}) + \beta \nabla f_{\xi_{t+1}}(x_{t+1}).$$

7: **end for**
8: **Output:** $\hat{x}_T \sim \text{Unif}\{x_0, \ldots, x_{T-1}\}$.

---

### 1.3. Theoretical Limitations of MARS

Yuan et al. (2025) study a variant of MARS (Algorithm 1) incorporating gradient clipping and a Hessian conditioning

matrix. Furthermore, they show that introducing the scaling coefficient $\gamma$ leads to gradient estimators with strictly smaller variance than those produced by MVR. However, their theoretical analysis does not distill this favorable property of the MARS estimator into an explicit convergence advantage of MARS over MVR. More precisely, Yuan et al. (2025) establish $\mathcal{O}(1/T^{2/3})$ rate bounds for $\mathbb{E}\|\nabla f(x_t) - g_t\|^2$ and $\mathbb{E}\|x_{t+1} - x_t\|^2$. Subsequent work by Liu et al. (2025) shows that MARS-M, which adapts MARS to the Muon optimizer for LLM training, achieves an $\mathcal{O}(1/T^{1/3})$ convergence rate in the gradient norm. Nevertheless, since MVR attains the same $\mathcal{O}(1/T^{1/3})$ rate, these results do not distinguish the theoretical performance of MARS from that of MVR, and therefore do not explain the consistently superior empirical behavior of MARS over MVR. Although Chang et al. (2025) consider MARS-M, their analysis focuses solely on the special case $\gamma = 1$, corresponding to MVR within the Muon framework, which has been recently studied by Huang et al. (2025); Khirirat et al. (2025); Qian et al. (2025).

Furthermore, the convergence guarantees for MARS by Yuan et al. (2025) and for MARS-M by Liu et al. (2025) rely on impractical, time-varying parameters. For instance, Theorem B.5 of Yuan et al. (2025) requires the scaling factor $\gamma_t \geq 0$ to depend on inaccessible quantities in practice:

$$\gamma_t = \frac{\|d_t\|^2 - G_t}{\beta_t \|d_t\|^2},$$

where $G_t = (1 - \beta_t)\mathbb{E}\langle \Delta_t, \nabla f_{\xi_t}(x_t) - \nabla f(x_t)\rangle + \beta_t \mathbb{E}\langle d_t, g_{t-1} - \nabla f(x_{t-1})\rangle$, and $d_t = \nabla f(x_t) - \nabla f(x_{t-1})$. The requirement of computing stochastic and full gradients at each iteration renders the implementation of their theoretical scaling factor $\gamma_t$ impossible. This highlights the need for a new theoretical framework that provides sharp provable guarantees for MARS, and explains its empirical superiority over MVR through improved complexity guarantees.

**A single-gradient variant of MARS.** The MARS momentum update in (4) requires evaluating two stochastic gradients per iteration, namely $\nabla f_{\xi_t}(x_t)$ and $\nabla f_{\xi_t}(x_{t-1})$. To reduce the per-iteration computational cost, one may instead form the correction term using $\nabla f_{\xi_t}(x_t) - \nabla f_{\xi_{t-1}}(x_{t-1})$, reusing the stochastic gradient computed at the previous iteration. However, in this paper, we focus on the two-gradient variant of MARS rather than its single-gradient variant. This is because the former achieves the $\mathcal{O}(1/T^{1/3})$ rate, which improves upon the $\mathcal{O}(1/T^{1/4})$ rate attained by the latter.

## 2. Contributions

The goal of this paper is to provide a theoretical explanation of why MARS empirically outperforms MVR without the

need to invoke any non-standard assumptions. Our key contributions include:

- **A refined similarity condition: $\gamma$-similarity.** To resolve this issue, we propose a more refined similarity notion called $\gamma$-similarity, which measures, in expectation, how well $\gamma(\nabla f_\xi(x) - \nabla f_\xi(y))$ tracks $\nabla f(x) - \nabla f(y)$ in the squared Euclidean norm. It provides a unified framework that recovers two widely used assumptions for analyzing stochastic optimization methods: (1) standard similarity ($\gamma = 1$) and (2) smoothness ($\gamma = 0$) as specific cases. Furthermore, under appropriate choices of $\gamma$, our $\gamma$-similarity constant is proved to be strictly smaller than the standard similarity and smoothness constants. This results in the conclusion that MARS theoretically outperforms MVR.

- **Superior convergence of MARS over MVR for nonconvex functions.** In Section 6, by using our $\gamma$-similarity condition, we derive a convergence guarantee for MARS under our $\gamma$-similarity condition and standard assumptions (smoothness and bounded variance). In contrast to Theorem B.5 of Yuan et al. (2025), which chooses $\gamma$ based on infeasible-to-compute quantities, our result holds for any fixed $\gamma \in [0, 1]$, and shows explicitly that MARS with a proper choice of $\gamma \in [0, 1]$ provides a strictly lower gradient complexity than standard MVR.

- **Stronger empirical performance of MARS over MVR and AdamW.** In Section 7.2, we benchmarked MARS, MVR, and AdamW for training a 124M parameter LLM model. Our experiments confirm the existence of an optimal $\gamma$ that maximizes token efficiency, outperforming both MVR and AdamW baselines.

## 3. Related Works

We review prior literature on variance reduction and stochastic momentum algorithms.

**Variance reduction.** While SGD is easy to implement and has been extensively studied, its stochastic gradient estimator suffers from high variance. To mitigate this issue, many variance reduction techniques have been proposed. Popular variance-reduction algorithms include SVRG (Johnson & Zhang, 2013), L-SVRG (Kovalev et al., 2020)), S2GD (Konečný & Richtárik, 2013), SAG (Roux et al., 2012; Schmidt et al., 2017), SAGA (Defazio et al., 2014a), FINITO (Defazio et al., 2014b), SPIDER (Fang et al., 2018), SARAH (Nguyen et al., 2017), and PAGE (Li et al., 2021). These algorithms construct variance-reduced stochastic gradient estimators by exploiting the difference between the

stochastic gradient and its full gradient. Despite many studies demonstrating the theoretical advantages of these variance reduction algorithms, they have limited success in training huge-scale neural network models. A key performance bottleneck is the need to compute the full gradient, which renders variance-reduction algorithms impractical for these tasks.

**Polyak momentum.** Inspired by Polyak's heavy-ball method (Polyak, 1964) for deterministic optimization, Polyak momentum is a widely used technique for improving the convergence of SGD. The convergence behaviors of SGD with momentum have been extensively studied under various settings (Yan et al., 2018; Yu et al., 2019; Gitman et al., 2019; Loizou & Richtárik, 2020; Liu et al., 2020; Sebbouh et al., 2021; Wang et al., 2023; Zhang et al., 2025; Oikonomou & Loizou, 2025). In particular, under fixed step-size and momentum parameters, Liu et al. (2020); Sebbouh et al. (2021) showed that SGD with momentum achieve convergence rates comparable to those of SGD when minimizing (strongly) convex functions. However, it often yields superior empirical performance, and is adopted as the default optimizers in open-source software libraries such as PyTorch (Paszke et al., 2019) and JAX (Bradbury et al., 2018).

**Novel momentum.** To improve upon Polyak momentum, several novel momentum techniques have been proposed. Notable techniques include Implicit Gradient Transport (IGT) (Arnold et al., 2019), Momentum Variance Reduction (MVR) (Cutkosky & Orabona, 2019), and various second-order momentum methods (Salehkaleybar et al., 2024; Tran & Cutkosky, 2022). In the context of non-convex stochastic optimization, while IGT achieves an $\mathcal{O}(1/T^{2/7})$ convergence rate in gradient norm (Cutkosky & Mehta, 2020), both MVR and second-order momentum reach $\mathcal{O}(1/T^{1/3})$, nearly matching the theoretical lower bounds established by Arjevani et al. (2023) and recently tightened by Fradin et al. (2026). Yuan et al. (2025) introduced MARS to improve the empirical performance of MVR. This approach was further extended by Liu et al. (2025) through the development of MARS-M, which adapts the MARS update for use within the Muon optimizer in LLM training. However, the existing theoretical analysis of MARS-M yields an $\mathcal{O}(1/T^{1/3})$ convergence rate. This rate is identical to the rates derived for MVR-based Muon variants (Chang et al., 2025; Huang et al., 2025; Khirirat et al., 2025; Qian et al., 2025). Consequently, despite its clear empirical advantages, a theoretical framework that explicitly demonstrates the superior convergence of MARS over MVR remains a significant gap in the literature.

## 4. Notations and Assumptions

We introduce notations and assumptions used throughout this paper.

### 4.1. Notations

We denote by $\nabla f_\xi$ the stochastic gradient associated with a random variable $\xi$, and by $\nabla f$ the full gradient. Expectation is denoted by $\mathbb{E}[\cdot]$, while $\mathbb{E}_\xi[\cdot]$ specifies expectation with respect to the randomness of $\xi$. Let $\hat{x}_T \sim \text{Unif}\{x_0, \ldots, x_{T-1}\}$ denote an iterate sampled uniformly at random from the sequence $\{x_0, x_1, \ldots, x_{T-1}\}$. For vectors $x, y \in \mathbb{R}^d$, $\langle x, y \rangle$ denotes their inner product, and $\|x\| := \sqrt{\langle x, x \rangle}$ the associated Euclidean norm. We use $e_i \in \mathbb{R}^d$ to denote the $i$th standard basis vector, $\text{Diag}(a_1, \ldots, a_d)$ the diagonal matrix with entries $a_1, \ldots, a_d \in \mathbb{R}$, and $I_d$ the $d \times d$ identity matrix. Finally, for a square matrix $A \in \mathbb{R}^{d \times d}$, $\lambda_{\max}(A)$ denotes its largest eigenvalue.

### 4.2. Assumptions

To facilitate our analysis, we impose assumptions on objectives and stochastic gradients, which are standard for analyzing stochastic algorithms for non-convex stochastic problems.

**Assumption 1** (Smoothness). The function $f : \mathbb{R}^d \to \mathbb{R}$ is bounded from below, i.e., $f_{\inf} := \inf_{x \in \mathbb{R}^d} f(x) > -\infty$, and is $L$-smooth, i.e.,

$$\|\nabla f(x) - \nabla f(y)\| \leq L\|x - y\|, \qquad \forall x, y \in \mathbb{R}^d.$$

$L$-Smoothness modulus of $f$ can be defined as

$$L := \sup_{x \neq y} \frac{\|\nabla f(x) - \nabla f(y)\|}{\|x - y\|} < \infty.$$

**Assumption 2** (Unbiased and variance-bounded stochastic gradients). The estimator $\nabla f_\xi(x)$ is unbiased of the gradient $\nabla f(x)$ and its variance is bounded, i.e., for all $x \in \mathbb{R}^d$,

$$\mathbb{E}_\xi \left[\nabla f_\xi(x)\right] = \nabla f(x), \quad \text{and}$$
$$\mathbb{E}_\xi \|\nabla f_\xi(x) - \nabla f(x)\|^2 \leq \sigma^2.$$

## 5. Similarity

We can measure how similar the stochastic gradient $\nabla f_\xi(x)$ is to the full gradient $\nabla f(x)$ by the following measure:

**Definition 1** ($\gamma$-similarity). For a fixed $\gamma \in \mathbb{R}$, the $\gamma$-similarity is

$$\delta_\gamma^2 := \sup_{x \neq y} \frac{\mathbb{E}_\xi \left[\|\gamma d_\xi(x, y) - d(x, y)\|^2\right]}{\|x - y\|^2}, \qquad (5)$$

where $d_\xi(x, y) = \nabla f_\xi(x) - \nabla f_\xi(y)$ and $d(x, y) = \nabla f(x) - \nabla f(y)$. We call $\delta_\gamma$ the $\gamma$-similarity constant whenever the above supremum is finite. Equivalently, for all $x, y \in \mathbb{R}^d$

$$\mathbb{E}_\xi \|\gamma(\nabla f_\xi(x) - \nabla f_\xi(y)) - (\nabla f(x) - \nabla f(y))\|^2 \leq \delta_\gamma^2 \|x - y\|^2.$$

Definition 1, which we call a $\gamma$-*similarity* constant, provides a unified framework that recovers constants related to several standard assumptions for analyzing stochastic optimization methods. When $\gamma = 0$, Definition 1 recovers smoothness in the sense that

$$\delta_0^2 = \sup_{x \neq y} \frac{\|d(x, y)\|^2}{\|x - y\|^2} \leq L^2.$$

If $L$ is chosen as the smallest valid smoothness constant, equality holds.

Conversely, by setting $\gamma = 1$, Definition 1 obtains the constant of a standard similarity condition (i.e., Lipschitz continuity of $\nabla f_\xi(x) - \nabla f(x)$ in expectation), since:

$$\delta^2 := \delta_1^2 = \sup_{x \neq y} \frac{\mathbb{E}_\xi \left[\|d_\xi(x, y) - d(x, y)\|^2\right]}{\|x - y\|^2}. \qquad (6)$$

The standard similarity condition with $\delta \geq 0$ in (6) is used to demonstrate improved convergence results for stochastic algorithms in both centralized (Tyurin et al., 2022; Chayti & Karimireddy, 2024) and distributed settings (Khaled & Jin, 2023; Karagulyan et al., 2024; Takezawa et al., 2025). This condition is more general than Assumption 2 of Karagulyan et al. (2024), a *star similarity* condition in Tovmasyan et al. (2026), and an *expected similarity* condition in Sadiev et al. (2024). If $\mathbb{E}_\xi \|\nabla f_\xi(x) - \nabla f_\xi(y)\|^2 \leq \hat{L}^2 \|x - y\|^2$ for some $\hat{L} > 0$, then it follows that $\delta \leq \hat{L}$.

For finite-sum minimization problems, or equivalently Problem (1) with $\xi$ being sampled uniformly at random from $\{1, 2, \ldots, n\}$, the standard similarity condition in (6) reduces to *second-order similarity* (Mairal, 2015; Khaled & Jin, 2023; Chayti & Karimireddy, 2024; Takezawa et al., 2025; Gasanov & Richtarik, 2024), i.e.

$$\frac{1}{n} \sum_{i=1}^n \|d_i(x, y) - d(x, y)\|^2 \leq \delta^2 \|x - y\|^2, \qquad (7)$$

which is more relaxed than Assumption 3.2. in Condat et al. (2025).

The next lemma provides the upper-bound of $\delta_\gamma^2$ according to Definition 1, based on the knowledge of $\delta^2$ and $L^2$.

**Lemma 1.** *Consider Problem* (1)*, and let $\delta_\gamma^2$ be defined in* (5)*. Suppose that Assumptions 1, 2 hold, and let $\delta^2 := \delta_1^2$ be the standard similarity modulus in* (6)*. Then:*

- *For any $\gamma \in \mathbb{R}$,*

$$\delta_\gamma^2 \leq \gamma^2 \delta^2 + (\gamma - 1)^2 L^2.$$

- *The minimizer of the right-hand side is*

$$\gamma_\star := \arg\min_{\gamma \in \mathbb{R}} \left\{ \gamma^2 \delta^2 + (\gamma - 1)^2 L^2 \right\} = \frac{L^2}{\delta^2 + L^2},$$

*which lies in the interval $[0, 1]$. Furthermore*

$$\delta_{\gamma_\star}^2 \leq \frac{L^2 \delta^2}{\delta^2 + L^2}.$$

Lemma 1 establishes that the $\gamma$-similarity constant $\delta_\gamma^2$ is upper-bounded by a combination of the standard similarity constant $\delta^2$ and the smoothness constant $L^2$. By selecting the optimal $\gamma_\star$ according to this lemma, we show that $\delta_{\gamma_\star}^2$ strictly improves upon both $\delta^2$ and $L^2$. In the case of quadratic problems, this upper bound becomes tight, and also $\delta_{\gamma_\star}^2$ can be substantially lower than its standard counterpart $\delta^2$.

**Example 1.** *Consider the problem of minimizing $f(x) = \frac{1}{n}\sum_{i=1}^n f_i(x)$ with $f_i(x) = \frac{1}{2}x^T A_i x$ for $x \in \mathbb{R}^d, d = n$ and $A_i = \hat{L}e_i e_i^T \in \mathbb{R}^{d \times d}$ for $\hat{L} > 0$ and $i$ being selected uniformly at random from $\{1, 2, \ldots, n\}$. Then, $\delta_\gamma^2 = (\gamma^2 n - 2\gamma + 1)\frac{\hat{L}^2}{n^2}$, $\delta^2 = (n-1)\frac{\hat{L}^2}{n^2}$, and the $L$-smoothness is $L^2 = \frac{\hat{L}^2}{n^2}$. This implies*

$$\delta_\gamma^2 = \gamma^2 \delta^2 + (\gamma - 1)^2 L^2.$$

*Also, $\gamma_\star := \underset{\gamma}{argmin}\ \gamma^2 \delta^2 + (\gamma - 1)^2 L^2 = \frac{1}{n}$ and $\delta_{\gamma_\star}^2 = \frac{n-1}{n}\frac{\hat{L}^2}{n^2}$, which yields $\frac{\delta^2}{\delta_{\gamma_\star}^2} = n$.*

# 6. Convergence of MARS

In this section, we establish the gradient complexity of MARS. In particular, we rely on the $\gamma$-similarity condition to explicitly prove that MARS attains a lower complexity than MVR.

To this end, we provide the convergence theorem for MARS to minimize nonconvex, smooth functions in the next theorem.

**Theorem 1.** *Consider MARS (Algorithm 1) for solving Problem (1). Suppose that Assumptions 1, 2 hold, and let $\delta_\gamma$ be the $\gamma$-similarity from Definition 1. Initialize $g_0 = \frac{1}{B_{\text{init}}}\sum_{j=1}^{B_{\text{init}}} \nabla f_{\xi_j}(x_0)$, and $B_{\text{init}} = \lceil 1/\beta \rceil$. For $\sigma = 0$ set $\beta = 1$, and for $\sigma > 0$, set*

$$\beta = \min\left\{1, \frac{\epsilon^2}{\sigma^2}\right\} \quad \text{and} \quad \eta = \frac{1}{L + \sqrt{a}},$$

*where $a = \left(\frac{(1-\beta)^3 |\gamma - 1|^2}{\beta}L^2 + 2(1-\beta)^2 \delta_\gamma^2\right)\frac{1}{\beta}$. Then, for $T \geq \left\lceil \frac{2\Delta}{\eta\epsilon^2} + \frac{\sigma^2}{\epsilon^2} \right\rceil$, for $\Delta = f(x_0) - f_{\inf}$, the output*

$\hat{x}_T \sim \text{Unif}\{x_0, \ldots, x_{T-1}\}$ *satisfies $\mathbb{E}\|\nabla f(\hat{x}_T)\|^2 \leq 4\epsilon^2$. Consequently, the number of stochastic gradient evaluations is*

$$\mathcal{O}\left(\frac{\sigma^2}{\epsilon^2} + \frac{L\Delta}{\epsilon^2} + \frac{\delta_\gamma \Delta \sigma}{\epsilon^3} + \frac{|\gamma - 1|L\Delta\sigma^2}{\epsilon^4}\right). \quad (8)$$

Theorem 1 generalizes the result by Yuan et al. (2025), in terms of the valid range for the coefficient $\gamma$ and gradient complexity bound. First, Theorem 1 holds for any $\gamma$-values, whereas Yuan et al. (2025) require specific $\gamma$ choices that depend on quantities typically unavailable in practice. Second, we provide explicit gradient complexity bounds, while Yuan et al. (2025) can establish only the rate bounds for $\mathbb{E}\|\nabla f(x_t) - g_t\|^2$ and $\mathbb{E}\|x_{t+1} - x_t\|^2$. Third, in contrast to Yuan et al. (2025), our theorem recovers the complexity bounds for MVR and SGD with momentum. Specifically, Theorem 1 obtains the $\mathcal{O}(1/\epsilon^3)$ complexity for MVR analyzed by Cutkosky & Orabona (2019); Fradin et al. (2026) when we let $\gamma = 1$, and the $\mathcal{O}(1/\epsilon^4)$ complexity for SGD with momentum.

When $\gamma = 1$, Theorem 1 recovers the MVR-type upper-bound scaling

$$\mathcal{O}\left(\frac{\sigma^2}{\epsilon^2} + \frac{L\Delta}{\epsilon^2} + \frac{\delta\Delta\sigma}{\epsilon^3}\right).$$

This is consistent with the known optimal-rate picture for stochastic nonconvex optimization under similarity-type conditions, including the bounds discussed by Fradin et al. (2026). We emphasize that our main contribution is not a new lower bound for $\gamma = 1$, but rather the refined $\gamma$-dependent upper bound showing when the scaled correction can improve the MVR bound expression.

In addition to the convergence for MARS under constant tuning parameters $\beta, \eta$, we also present the convergence for MARS under $T$-dependent parameters in the next corollary.

**Corollary 1** (A $T$-dependent parameter choice). *Consider MARS (Algorithm 1) for solving Problem (1), where Assumptions 1, 2 hold. Let $\delta_\gamma^2$ be defined by (5), and let the algorithm run for horizon $T \geq 1$, such that $\sqrt{\frac{|\gamma-1|L\Delta}{\sigma^2 T}} + 2^{-1/3}\left(\frac{\delta_\gamma\Delta}{\sigma^2 T}\right)^{2/3} \leq 1$, and choose*

$$\beta_T = \sqrt{\frac{|\gamma - 1|L\Delta}{\sigma^2 T}} + 2^{-1/3}\left(\frac{\delta_\gamma\Delta}{\sigma^2 T}\right)^{2/3},$$

$$\eta = \frac{1}{L + \frac{|\gamma-1|L}{\beta_T} + \frac{\sqrt{2}\,\delta_\gamma}{\sqrt{\beta_T}}}.$$

*Then for $g_0 = \frac{1}{B_{\text{init}}} \sum_{j=1}^{B_{\text{init}}} \nabla f_{\xi_j}(x_0)$ with $B_{\text{init}} = \lceil 1/\beta_T \rceil$,*

$$\mathbb{E}\left[\|\nabla f(\hat{x}_T)\|^2\right] \leq \frac{2L\Delta}{T} + \frac{4\sigma\sqrt{|\gamma - 1|\, L\, \Delta}}{\sqrt{T}} + \frac{3 \cdot 2^{2/3}(\delta_\gamma \Delta\sigma)^{2/3}}{T^{2/3}} + \frac{\sigma^2}{T}, \quad (9)$$

*where $\hat{x}_T \sim \text{Unif}\{x_0, \ldots, x_{T-1}\}$.*

Like Theorem 1, Corollary 1 encompasses the convergence rate bound for MVR and SGD with momentum, depending on the coefficient $\gamma$. On the one hand, when we let $\gamma = 1$, Corollary 1 obtains the $\mathcal{O}(T^{-2/3})$ rate in the squared gradient norm for MVR (up to the additional $\sigma^2/T$ term arising from initialization $\Delta$). On the other hand, when we let $\gamma = 0$, Corollary 1 yields the $\mathcal{O}(T^{-1/2})$ rate for SGD with momentum.

## 6.1. Theoretical Advantage of MARS over MVR

Next, we demonstrate how Theorem 1 implies the superior performance of MARS with $\gamma \in [0, 1]$ to MVR, in the next corollary.

**Corollary 2.** *Consider the setting of Theorem 1 with $\gamma \in$*

[0, 1]. *Let*

$$A = \frac{\Delta\sigma}{\epsilon^3}, \qquad B = \frac{L\Delta\sigma^2}{\epsilon^4}, \qquad D = \delta^2 + L^2.$$

*Define the surrogate non-common part of the MARS bound by $J(\gamma) = A\sqrt{\gamma^2\delta^2 + (1-\gamma)^2 L^2} + B(1-\gamma)$. If $B < A\delta$ and $\gamma_\star = \frac{L^2}{D} + \frac{BL\delta}{D\sqrt{A^2 D - B^2}}$, then $\gamma_\star \in [0, 1]$ and*

$$J(\gamma_\star) \leq J(1) = A\delta.$$

*Consequently, after replacing $\delta_\gamma$ by the upper bound from Lemma 1, MARS complexity bound in (8) is no larger than the corresponding displayed MVR bound at $\gamma = 1$.*

*Proof.* The terms common to the displayed MARS and MVR bounds are $\sigma^2/\epsilon^2$ and $L\Delta/\epsilon^2$. Hence it suffices to compare the remaining displayed terms. By Lemma 1, for any $\gamma \in [0, 1]$,

$$\frac{\Delta\sigma}{\epsilon^3}\delta_\gamma + \frac{L\Delta\sigma^2}{\epsilon^4}(1-\gamma)$$

is upper-bounded by $A\sqrt{\gamma^2\delta^2 + (1-\gamma)^2 L^2} + B(1-\gamma) = J(\gamma)$. Proposition 1 gives $J(\gamma_\star) \leq J(1) = A\delta$ whenever

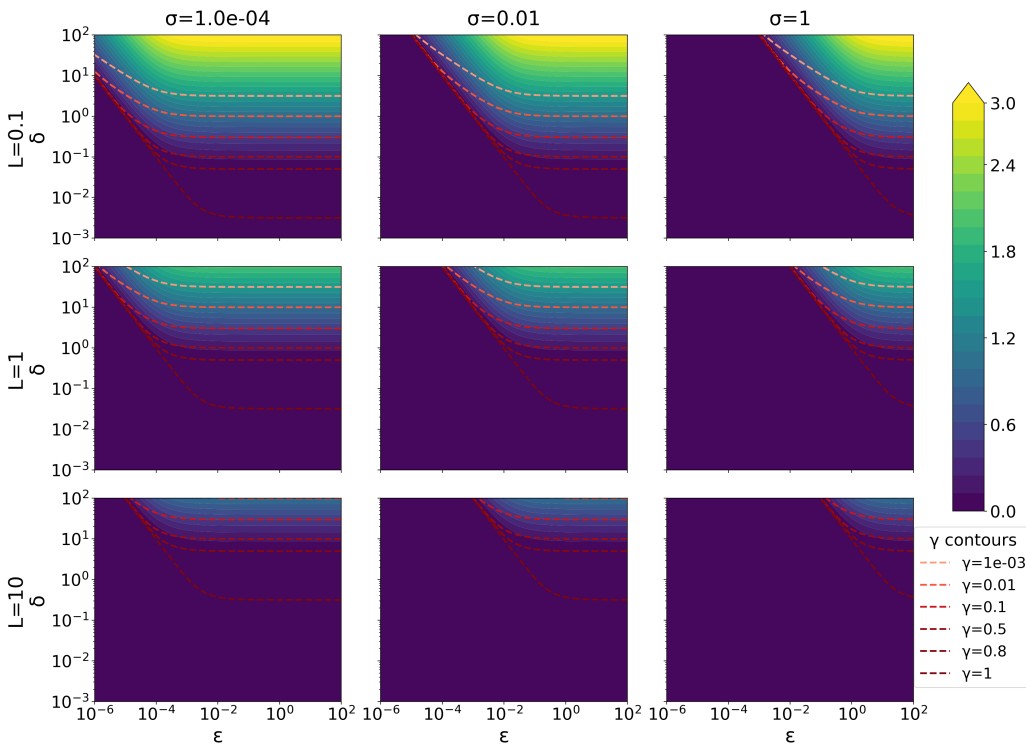

*Figure 1.* **Bound-implied MARS vs. MVR speedup.** Brighter colors indicate settings where MARS offers larger improvements over the MVR baseline ($\gamma = 1$) and dark regions show no benefit. Color encodes $\log_{10}(A\delta / J(\gamma_\star))$, where $J(\gamma) = A\sqrt{\gamma^2\delta^2 + (1-\gamma)^2 L^2} + B(1-\gamma)$ with $A = \sigma/\epsilon^3$ $B = L\sigma^2/\epsilon^4$, and $\gamma_\star$ is given in Corollary 2. Each panel sweeps over target accuracy $\epsilon$ (horizontal) and gradient heterogeneity $\delta$ (vertical). Rows vary smoothness $L \in \{0.1, 1, 10\}$, columns vary noise $\sigma \in \{10^{-4}, 10^{-2}, 1\}$. Dashed curves show optimal $\gamma_\star$ contours.

$B < A\delta$. Therefore the non-common part of the displayed MARS upper-bound expression at $\gamma_\star$ is no larger than the corresponding non-common part of the displayed MVR expression at $\gamma = 1$. □

This corollary shows that, under the stated condition and surrogate upper bound on $\delta_\gamma$, the displayed MARS upper-bound expression can be made no larger than the corresponding MVR expression.

### 6.2. Illustrative Speed-up Gains from MARS over MVR

The only $\gamma$-dependent part of the displayed bound in Theorem 1 is controlled by

$$A\delta_\gamma + B(1 - \gamma), \qquad A = \frac{\Delta\sigma}{\epsilon^3}, \qquad B = \frac{L\Delta\sigma^2}{\epsilon^4}.$$

Using Lemma 1, this is upper-bounded by the surrogate

$$J(\gamma) = A\sqrt{\gamma^2\delta^2 + (1 - \gamma)^2 L^2} + B(1 - \gamma).$$

Since the MVR baseline corresponds to $J(1) = A\delta$, we visualize the bound-implied speedup by the ratio $A\delta/J(\gamma_\star)$.

To illustrate the magnitude of the complexity improvement offered by MARS over MVR, Figure 1 displays the quantity $\log_{10}(A\delta/J(\gamma_\star))$ across various parameter settings. Specifically, we evaluate this speed-up for smoothness constants $L \in \{0.1, 1.0, 10.0\}$ and noise levels $\sigma \in \{10^{-4}, 10^{-2}, 1\}$.

In Figure 1, three patterns emerge. First, speedup grows toward larger $\epsilon$ and $\delta$: the upper-right of each panel is bright, reflecting regimes where the $(1 - \gamma)$-penalty becomes negligible relative to the baseline cost. The $\gamma_\star$ contours shift toward smaller values in this region, indicating that the surrogate increasingly prefers $\gamma_\star < 1$. Second, a diagonal transition separates improvement from no-improvement,

governed by the ratio $L\sigma/(\epsilon\delta)$: improvement emerges when $\epsilon\delta$ is sufficiently large relative to $L\sigma$. Third, increasing $L$ (top to bottom) or $\sigma$ (left to right) shifts the improvement region outward. Overall, MARS yields the largest gains in moderate-accuracy, high-heterogeneity regimes, while harder problems (large $L\sigma$) require proportionally larger $\epsilon\delta$ to benefit.

## 7. Experiments

### 7.1. A CIFAR-10 Probe of the Predicted Correction Scale

We first run a small theorem-aligned probe to connect the analysis to quantities that can be estimated along a training trajectory. This experiment is not intended as a competitive CIFAR-10 benchmark. Its purpose is to test whether the local gradient-difference statistics appearing in $\gamma$-similarity prefer a correction scale below the MVR value $\gamma = 1$.

We train a small CNN on CIFAR-10 using the vanilla two-gradient $\gamma$-MVR update analyzed in Theorem 1. At a checkpoint $t$, let

$$d_t = \nabla f(x_t) - \nabla f(x_{t-1}),$$
$$d_{B,t} = \nabla f_B(x_t) - \nabla f_B(x_{t-1}),$$

where $B$ is a mini-batch. Using $M = 512$ sampled mini-batches, we estimate the local predicted correction scale

$$\widehat{\gamma}_t^\star = \frac{\|d_t\|^2}{\frac{1}{M}\sum_{m=1}^M \|d_{B_m,t}\|^2},$$

which is the empirical analogue of the minimizer suggested by Lemma 1. We also compute the grid minimizer of the corresponding local correction proxy over the tested $\gamma$ values.

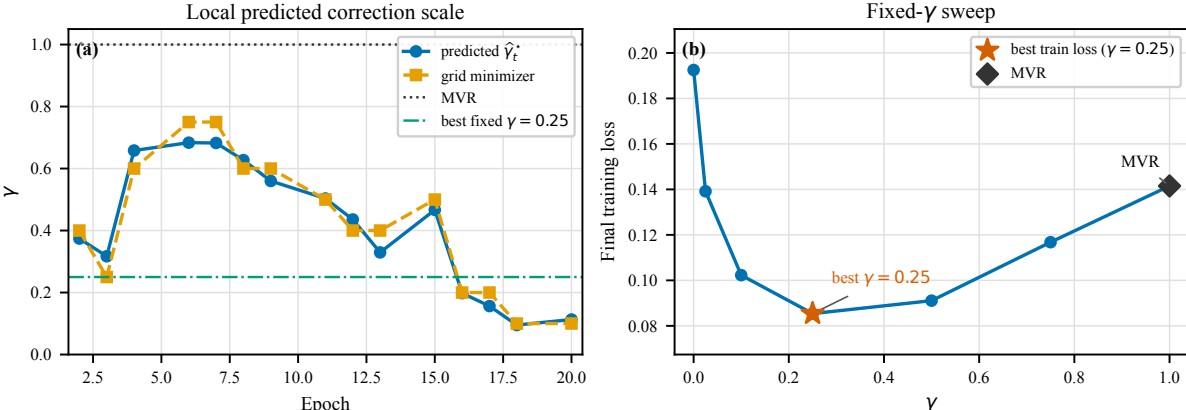

*Figure 2.* **CIFAR-10 probe of the MARS correction scale.** Left: checkpoint-level predicted scale $\widehat{\gamma}_t^\star = \|d_t\|^2/(\frac{1}{M}\sum_m \|d_{B_m,t}\|^2)$ and the grid minimizer of the local correction proxy. Both remain below the MVR value $\gamma = 1$ and vary during training. Right: fixed-$\gamma$ sweep in the same vanilla two-gradient $\gamma$-MVR setup. Final training loss is minimized at $\gamma < 1$, while the MVR setting $\gamma = 1$ is worse under the same hyperparameters.

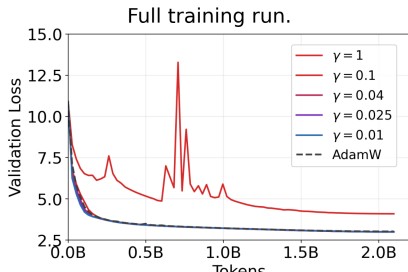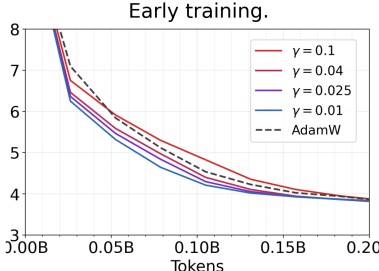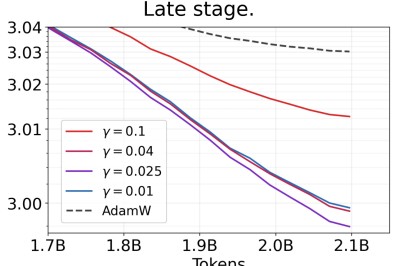

*Figure 3.* **Validation loss vs. tokens for a 124M $\gamma$-sweep.** Left: full training run. Center: early training (first $\approx 0.2$B tokens). Right: late stage near the Chinchilla-optimal budget (log-scale).

Figure 2 shows that the predicted correction scale stays below 1 throughout training and changes substantially over the trajectory. In this run, $\widehat{\gamma}_t^\star$ ranges from approximately 0.10 to 0.68, with median about 0.44. The grid minimizer follows the same qualitative pattern and is also consistently below 1. A fixed-$\gamma$ sweep in the same setup further shows that final training loss is minimized at $\gamma = 0.25$, whereas the MVR setting $\gamma = 1$ is worse under the same hyperparameters. These results support the qualitative message of the theory: the useful correction scale depends on local gradient-difference statistics and need not coincide with MVR.

### 7.2. MARS-AdamW on GPT-style Pretraining

We evaluate the effect of $\gamma$ in MARS-AdamW in an applied setting, since LLM pretraining is a primary modern workload. Concretely, we follow the pretraining protocol and codebase of Semenov et al. (2025) on a 124M Llama-style model (12L/12H/768) and a Chinchilla-optimal budget (Hoffmann et al., 2022) of approximately 2.10B tokens; the protocol is well-tuned, so we rely on their hyperparameters rather than re-tuning for each $\gamma$. Our goal is not exhaustive hyperparameter search, but to observe the behavior of different $\gamma$ values under a sane and inexpensive setup; we sweep $\gamma \in \{0.01, 0.025, 0.04, 0.1, 1\}$ and include AdamW as a baseline. We keep all hyperparameters fixed across $\gamma$ to isolate the effect of the MARS scaling.

**Results.** Figure 3 shows three views of the same training run: the full trajectory (left), an early-stage zoom (center), and a late-stage zoom (right). Overall, we observe three phenomena. First, small $\gamma$ values can outperform AdamW; the late-stage panel (right) shows several $\gamma < 1$ curves below AdamW near the end of training. Second, different small $\gamma$ values behave differently and there is an optimal choice; the early-stage ordering (center) differs from the late-stage ordering (right), which is consistent with the iterates traversing regions with different local properties that change

how $\gamma$ trades off the relevant terms in our bound (e.g., the $\delta_\gamma$-dependent term versus the $(1 - \gamma)$-dependent penalty), though the final-loss differences among small $\gamma$ values are relatively small. Third, the classic MVR setting ($\gamma = 1$) can be sensitive to hyperparameters and exhibit instability without dedicated tuning, as visible in the full-trajectory panel (left). Additional experimental details are provided in Appendix G.

## 8. Conclusion

In this paper, we have provided a rigorous theoretical explanation for the superior convergence performance of MARS over MVR algorithms. By introducing the $\gamma$-similarity measure, we derive convergence guarantees of MARS for minimizing nonconvex functions solely under assumptions commonly used for analyzing stochastic algorithms. Our results prove that MARS with an appropriately tuned $\gamma \in [0, 1]$ is explicitly shown to achieve a strictly lower gradient complexity than MVR. This theoretical framework is corroborated by our empirical studies on GPT pretraining, which show that there exists an optimal choice of $\gamma$ maximizing token efficiency of MARS over both MVR and AdamW.

## Acknowledgements

The research reported in this publication was supported by funding from King Abdullah University of Science and Technology (KAUST): i) KAUST Baseline Research Scheme, ii) Center of Excellence for Generative AI, under award number 5940, iii) SDAIA-KAUST Center of Excellence in Artificial Intelligence and Data Science.

## Impact Statement

Our paper introduces the $\gamma$-similarity condition, a novel tool to better capture the nuances of specific algorithmic modifications. Using this condition, we provide the first theoretical

justification for the superior performance of MARS. This will pave the way for novel analysis frameworks for scaled momentum and variance reduction methods.

Furthermore, our theoretical findings show how the optimal scaled coefficient $\gamma$ improves efficiency in large-scale model training. This will lead to the development of scaled momentum and variance reduction methods, which reduce computational costs and improve resource efficiency in training state-of-the-art learning models.

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

# Contents

## A. Proof of Lemma 1

Fix $x \neq y$ and write

$$d_\xi = d_\xi(x, y), \qquad d = d(x, y).$$

By Assumption 2, $\mathbb{E}_\xi[d_\xi] = d$, and hence $\mathbb{E}_\xi[d_\xi - d] = 0$. Therefore,

$$
\begin{aligned}
\mathbb{E}_\xi \|\gamma d_\xi - d\|^2 &= \mathbb{E}_\xi \|\gamma(d_\xi - d) + (\gamma - 1)d\|^2 \\
&= \gamma^2 \mathbb{E}_\xi \|d_\xi - d\|^2 + (\gamma - 1)^2 \|d\|^2 + 2\gamma(\gamma - 1) \langle \mathbb{E}_\xi[d_\xi - d], d \rangle \\
&= \gamma^2 \mathbb{E}_\xi \|d_\xi - d\|^2 + (\gamma - 1)^2 \|d\|^2.
\end{aligned}
$$

Dividing by $\|x - y\|^2$ and using the definitions of $\delta$ and $L$ gives

$$\frac{\mathbb{E}_\xi \|\gamma d_\xi(x, y) - d(x, y)\|^2}{\|x - y\|^2} \leq \gamma^2 \delta^2 + (\gamma - 1)^2 L^2.$$

Taking the supremum over $x \neq y$ proves the first claim.

The function $q(\gamma) = \gamma^2 \delta^2 + (\gamma - 1)^2 L^2$ is a convex quadratic, and $q'(\gamma) = 2\gamma\delta^2 + 2(\gamma - 1)L^2$. Solving $q'(\gamma) = 0$ yields

$$\gamma_\star = \frac{L^2}{\delta^2 + L^2} \in [0, 1].$$

Substituting this value gives

$$q(\gamma_\star) = \frac{L^2 \delta^2}{\delta^2 + L^2}.$$

Since $\delta_{\gamma_\star}^2 \leq q(\gamma_\star)$, the proof is complete.

## B. Additional Lemma for $\gamma$-Similarity

In this section, we include one additional lemma, which characterizes the property of $\gamma$-similarity.

**Lemma 2.** *Under Assumption 2, consider $h(x, y; \gamma) = \mathbb{E}_\xi \|\gamma d_\xi(x, y) - d(x, y)\|^2$, where $d_\xi(x, y) = \nabla f_\xi(x) - \nabla f_\xi(y)$ and $d(x, y) = \nabla f(x) - \nabla f(y)$. Then, for a fixed $x, y \in \mathbb{R}^d$,*

$$h(x, y; \gamma) = \gamma^2 \mathbb{E}_\xi \|d_\xi(x, y) - d(x, y)\|^2 + (\gamma - 1)^2 \|d(x, y)\|^2.$$

*In addition,*

1. *$h(x, y; \gamma)$ is convex with respect to $\gamma$.*

2. *$h(x, y; \gamma)$ is non-increasing for $\gamma < 0$, and is non-decreasing for $\gamma > 1$.*

3. If $\mathbb{E}_\xi \|d_\xi(x,y)\|^2 > 0$, then the minimizer $\gamma_\star := \operatorname{argmin}_{\gamma \in \mathbb{R}} h(x,y;\gamma)$ is

$$\gamma_\star = \frac{\|d(x,y)\|^2}{\mathbb{E}_\xi \|d_\xi(x,y) - d(x,y)\|^2 + \|d(x,y)\|^2} = \frac{\|d(x,y)\|^2}{\mathbb{E}_\xi \|d_\xi(x,y)\|^2} \in [0,1],$$

*and*

$$h(x,y;\gamma_\star) = \gamma_\star \mathbb{E}_\xi \|d_\xi(x,y) - d(x,y)\|^2 = (1-\gamma_\star)\|d(x,y)\|^2.$$

*If $\mathbb{E}_\xi \|d_\xi(x,y)\|^2 = 0$, then $h(x,y;\gamma) = 0$ for all $\gamma$, and every $\gamma \in \mathbb{R}$ is a minimizer.*

## B.1. Proof of Lemma 2

We begin by proving the first statement. From the definition of $h(x,y;\gamma)$, and the Euclidean norm, for any fixed $x,y \in \mathbb{R}^d$,

$$
\begin{aligned}
h(x,y;\gamma) &= \mathbb{E}_\xi \|\gamma(d_\xi(x,y) - d(x,y)) + (\gamma-1)d(x,y)\|^2 \\
&= \gamma^2 \mathbb{E}_\xi \|d_\xi(x,y) - d(x,y)\|^2 + (\gamma-1)^2 \mathbb{E}_\xi \|d(x,y)\|^2 + 2\gamma(\gamma-1)\mathbb{E}_\xi \langle d_\xi(x,y) - d(x,y), d(x,y) \rangle \\
&= \gamma^2 \mathbb{E}_\xi \|d_\xi(x,y) - d(x,y)\|^2 + (\gamma-1)^2 \mathbb{E}_\xi \|d(x,y)\|^2 + 2\gamma(\gamma-1)\langle \mathbb{E}_\xi[d_\xi(x,y) - d(x,y)], d(x,y) \rangle.
\end{aligned}
$$

By Assumption 2, i.e. $\mathbb{E}_\xi[d_\xi(x,y) - d(x,y)] = 0$, and by the fact that $\mathbb{E}_\xi \|d(x,y)\|^2 = \|d(x,y)\|^2$,

$$h(x,y;\gamma) = \gamma^2 \mathbb{E}_\xi \|d_\xi(x,y) - d(x,y)\|^2 + (\gamma-1)^2 \|d(x,y)\|^2.$$

Next, we prove the first statement. By taking the second derivative with respect to $\gamma$,

$$\frac{d^2}{d\gamma^2} h(x,y;\gamma) = 2\mathbb{E}_\xi \|d_\xi(x,y) - d(x,y)\|^2 + 2\|d(x,y)\|^2.$$

Since $\frac{d^2}{d\gamma^2} h(x,y;\gamma) \geq 0$ for any $\gamma \in \mathbb{R}$, $h(x,y;\gamma)$ is a convex function.

Next, we prove the second statement. By taking the first derivative with respect to $\gamma$,

$$\frac{d}{d\gamma} h(x,y;\gamma) = 2\gamma \mathbb{E}_\xi \|d_\xi(x,y) - d(x,y)\|^2 + 2(\gamma-1)\|d(x,y)\|^2.$$

If $\gamma < 0$, then $\frac{d}{d\gamma} h(x,y;\gamma) < 0$, thus implying that $h(x,y;\gamma)$ is non-increasing.

If $\gamma > 1$, then $\frac{d}{d\gamma} h(x,y;\gamma) > 0$, thus meaning that $h(x,y;\gamma)$ is non-decreasing.

Next, we prove the third statement. As $h(x,y;\gamma)$ is convex, we can find $\gamma_\star := \operatorname{argmin}_\gamma h(x,y;\gamma)$ by solving

$$\frac{d}{d\gamma} h(x,y;\gamma) = 0.$$

This implies that

$$\gamma_\star = \frac{\|d(x,y)\|^2}{\mathbb{E}_\xi \|d_\xi(x,y) - d(x,y)\|^2 + \|d(x,y)\|^2} \in [0,1],$$

provided that the denominator is nonzero. Substituting this value into $h$ gives $h(x,y;\gamma_\star) = \gamma_\star \mathbb{E}_\xi \|d_\xi(x,y) - d(x,y)\|^2$. Furthermore, since $\mathbb{E}_\xi d_\xi(x,y) = d(x,y)$ and $\mathbb{E}_\xi \|d_\xi(x,y) - d(x,y)\|^2 = \mathbb{E}_\xi \|d_\xi(x,y)\|^2 - \|d(x,y)\|^2$, we can show that

$$h(x,y;\gamma) = \gamma^2 \mathbb{E}_\xi \|d_\xi(x,y)\|^2 + (1-2\gamma)\|d(x,y)\|^2,$$

and that

$$\gamma_\star = \frac{\|d(x,y)\|^2}{\mathbb{E}_\xi \|d_\xi(x,y)\|^2},$$

which yields $h(x,y;\gamma_\star) = (1-\gamma_\star)\|d(x,y)\|^2$.

## B.2. Proof of Example 1

Consider the problem of minimizing $f(x) = \frac{1}{n}\sum_{i=1}^{n} f_i(x)$ with each component function $f_i(x) = \frac{1}{2}x^T A_i x$ for $x \in \mathbb{R}^d$ and $A_i = \hat{L}e_i e_i^T \in \mathbb{R}^{d \times d}$ for $\hat{L} > 0$ and $i$ being selected uniformly at random from $\{1, 2, \ldots, n\}$. Then,

$$
\mathbb{E}_i \left\| \gamma d_i(x, y) - d(x, y) \right\|^2
$$
$$
= \mathbb{E}_i \left\| (\gamma A_i - \bar{A})(x - y) \right\|^2
$$
$$
= \frac{1}{n}\sum_{i=1}^{n} \left\| (\gamma A_i - \bar{A})(x - y) \right\|^2
$$
$$
= (x - y)^T \left( \frac{1}{n}\sum_{i=1}^{n}(\gamma A_i - \bar{A})^T (\gamma A_i - \bar{A}) \right)(x - y)
$$
$$
\leq \lambda_{\max}\left( \frac{1}{n}\sum_{i=1}^{n}(\gamma A_i - \bar{A})^T (\gamma A_i - \bar{A}) \right) \left\| x - y \right\|^2,
$$

where $\bar{A} = \frac{1}{n}\sum_{i=1}^{n} A_i = \frac{\hat{L}}{n}I_n$.

Next, by the definition of $A_i$ and $\bar{A}$, we can show that

$$
\gamma A_i - \bar{A}
$$
$$
= \mathrm{Diag}\left( -\frac{\hat{L}}{n}, \ldots, -\frac{\hat{L}}{n}, \underbrace{\gamma\hat{L} - \frac{\hat{L}}{n}}_{i^{\text{th}}-\text{coordinate}}, -\frac{\hat{L}}{n}, \ldots, -\frac{\hat{L}}{n} \right),
$$

and that

$$
\lambda_{\max}\left( \frac{1}{n}\sum_{i=1}^{n}(\gamma A_i - \bar{A})^T (\gamma A_i - \bar{A}) \right)
$$
$$
= \frac{1}{n}\left[ \left( \gamma\hat{L} - \frac{\hat{L}}{n} \right)^2 + (n-1)\frac{\hat{L}^2}{n^2} \right]
$$
$$
= (\gamma^2 n - 2\gamma + 1)\frac{\hat{L}^2}{n^2}.
$$

Therefore,

$$
\mathbb{E}_i \left\| \gamma d_i(x, y) - d(x, y) \right\|^2 \leq \delta_\gamma^2 \left\| x - y \right\|^2, \tag{10}
$$

where $\delta_\gamma^2 = \left[ (\gamma^2 n - 2\gamma + 1)\frac{\hat{L}^2}{n^2} \right]$. This implies that the $\gamma$-similarity parameter (10) is $\delta_\gamma^2 = (\gamma^2 n - 2\gamma + 1)\frac{\hat{L}^2}{n^2}$, the standard similarity parameter ((10) with $\gamma = 1$) is $\delta^2 = (n-1)\frac{\hat{L}^2}{n^2}$, and the $L$-smoothness ((10) with $\gamma = 0$) is $L^2 = \frac{\hat{L}^2}{n^2}$.

In conclusion, $\delta_\gamma^2 = \gamma^2\delta^2 + (\gamma - 1)^2 L^2$, and

$$
\gamma_\star := \underset{\gamma}{\arg\min}\, \gamma^2\delta^2 + (\gamma - 1)^2 L^2 = \frac{1}{n},
$$

with its associated optimal value at $\delta_{\gamma_\star}^2 = \frac{n-1}{n}\frac{\hat{L}^2}{n^2}$. This implies that $\frac{\delta^2}{\delta_{\gamma_\star}^2} = n$.

# C. Descent Lemma for MARS Algorithms

We provide the descent inequality for MARS algorithms to solve stochastic optimization problems in (1).

We begin by introducing one useful lemma for deriving an easy-to-write stepsize bound.

**Lemma 3** (Lemma 5 of Richtárik et al. (2021))**.** *Let $a, b > 0$. If $0 < \eta \leq \frac{1}{\sqrt{a}+b}$, then $a\eta^2 + b\eta \leq 1$.*

Now, we provide the descent lemma below.

**Lemma 4.** *Consider* MARS *(Algorithm 1) for solving Problem (1), where Assumptions 1, 2 hold and let $\delta_\gamma$ be defined by (5). If*

$$0 < \eta \leq \frac{1}{\sqrt{a}+L}, \qquad a = \left( \frac{(1-\beta)^3|\gamma-1|^2}{\beta} L^2 + 2(1-\beta)^2\delta_\gamma^2 \right) \frac{1}{\beta},$$

*then*

$$\mathbb{E}[H_{t+1}] \leq \mathbb{E}[H_t] - \frac{\eta}{2}\mathbb{E}\|\nabla f(x_t)\|^2 + \beta\eta\sigma^2,$$

*where $H_t = f(x_t) - f_{\inf} + \frac{\eta}{2\beta}\|e_t\|^2$ and $e_t = g_t - \nabla f(x_t)$.*

*Proof.* Let $\mathcal{F}_t$ be the history up to iteration $t$, i.e. $\mathcal{F}_t := \sigma(x_0, g_0, \xi_1, \xi_2, \ldots, \xi_t)$. For $t \geq 1$, define the conditional expectation

$$\mathbb{E}_t[\cdot] := \mathbb{E}[\cdot \mid \mathcal{F}_{t-1}],$$

so the only randomness in $\mathbb{E}_t[\cdot]$ comes from the fresh sample $\xi_t$.

**Step 1) Error term.** Let $d_t = \nabla f_{\xi_t}(x_t) - \nabla f_{\xi_t}(x_{t-1})$, $D_t = \nabla f(x_t) - \nabla f(x_{t-1})$, and $z_t = \nabla f_{\xi_t}(x_t) - \nabla f(x_t)$. From the update for $g_t$,

$$\begin{aligned}
g_t &= (1-\beta)(g_{t-1} + \gamma d_t) + \beta\nabla f_{\xi_t}(x_t) \\
&= (1-\beta)(g_{t-1} + \gamma d_t - D_t) + (1-\beta)D_t + \beta\nabla f(x_t) + \beta z_t.
\end{aligned}$$

Define the error $e_t := g_t - \nabla f(x_t)$. Then,

$$\begin{aligned}
e_t &= (1-\beta)g_{t-1} + (1-\beta)(\gamma d_t - D_t) + (1-\beta)D_t - (1-\beta)\nabla f(x_t) + \beta z_t \\
&= (1-\beta)e_{t-1} + (1-\beta)(\gamma d_t - D_t) + \beta z_t \\
&= (1-\beta)e_{t-1} + w_t,
\end{aligned}$$

where $w_t := (1-\beta)(\gamma d_t - D_t) + \beta z_t$.

Taking conditional expectation $\mathbb{E}_t[\cdot]$ yields

$$\begin{aligned}
\mathbb{E}_t\|e_t\|^2 &= \mathbb{E}_t\|(1-\beta)e_{t-1} + w_t\|^2 \\
&= (1-\beta)^2\|e_{t-1}\|^2 + 2(1-\beta)\langle e_{t-1}, \mathbb{E}_t[w_t]\rangle + \mathbb{E}_t\|w_t\|^2,
\end{aligned}$$

since $e_{t-1}$ is $\mathcal{F}_{t-1}$-measurable.

Next, from Assumption 2 (unbiasedness),

$$\mathbb{E}_t[w_t] = (1-\beta)\mathbb{E}_t[\gamma d_t - D_t] + \beta\mathbb{E}_t[z_t] = (1-\beta)(\gamma-1)D_t.$$

Therefore,

$$\begin{aligned}
\mathbb{E}_t\|e_t\|^2 &= (1-\beta)^2\|e_{t-1}\|^2 + 2(1-\beta)^2(\gamma-1)\langle e_{t-1}, D_t\rangle + \mathbb{E}_t\|w_t\|^2 \\
&\leq (1-\beta)^2\|e_{t-1}\|^2 + 2(1-\beta)^2|\gamma-1|\,\|e_{t-1}\|\,\|D_t\| + \mathbb{E}_t\|w_t\|^2 \\
&\leq (1-\beta)^2(1+\theta)\|e_{t-1}\|^2 + \frac{(1-\beta)^2|\gamma-1|^2}{\theta}\|D_t\|^2 + \mathbb{E}_t\|w_t\|^2,
\end{aligned}$$

where we reach the first inequality by the Cauchy-Schwarz inequality, and the last inequality by the fact that $2ab \le \theta a^2 + \frac{1}{\theta}b^2$ with $\theta > 0$.

To complete the bound of $\mathbb{E}_t\|e_t\|^2$, we now bound $\mathbb{E}_t\|w_t\|^2$:

$$
\begin{aligned}
\mathbb{E}_t\|w_t\|^2 &= \mathbb{E}_t\|(1-\beta)(\gamma d_t - D_t) + \beta z_t\|^2 \\
&\le 2(1-\beta)^2\mathbb{E}_t\|\gamma d_t - D_t\|^2 + 2\beta^2\mathbb{E}_t\|z_t\|^2 \\
&\overset{(5)}{\le} 2(1-\beta)^2\delta_\gamma^2\|x_t - x_{t-1}\|^2 + 2\beta^2\mathbb{E}_t\|z_t\|^2 \\
&\overset{\text{Assumption 2}}{\le} 2(1-\beta)^2\delta_\gamma^2\|x_t - x_{t-1}\|^2 + 2\beta^2\sigma^2.
\end{aligned}
$$

Also, by $L$-smoothness of $f$ (Assumption 1),

$$
\|D_t\| \le L\|x_t - x_{t-1}\| \quad \Rightarrow \quad \|D_t\|^2 \le L^2\|x_t - x_{t-1}\|^2.
$$

Thus,

$$
\begin{aligned}
\mathbb{E}_t\|e_t\|^2 &\le (1-\beta)^2(1+\theta)\|e_{t-1}\|^2 \\
&\quad + \left(\frac{(1-\beta)^2|\gamma - 1|^2}{\theta}L^2 + 2(1-\beta)^2\delta_\gamma^2\right)\|x_t - x_{t-1}\|^2 + 2\beta^2\sigma^2.
\end{aligned}
$$

Taking full expectation and using $\mathbb{E}[\mathbb{E}_t[\cdot]] = \mathbb{E}[\cdot]$ gives

$$
\begin{aligned}
\mathbb{E}\|e_t\|^2 &\le (1-\beta)^2(1+\theta)\mathbb{E}\|e_{t-1}\|^2 \\
&\quad + \left(\frac{(1-\beta)^2|\gamma - 1|^2}{\theta}L^2 + 2(1-\beta)^2\delta_\gamma^2\right)\mathbb{E}\|x_t - x_{t-1}\|^2 + 2\beta^2\sigma^2.
\end{aligned}
$$

Choose $\theta = \frac{\beta}{1-\beta}$ (for $\beta \in (0, 1)$), so that $(1-\beta)^2(1+\theta) = 1 - \beta$ and $\frac{(1-\beta)^2}{\theta} = \frac{(1-\beta)^3}{\beta}$, yielding

$$
\mathbb{E}\|e_t\|^2 \le (1-\beta)\mathbb{E}\|e_{t-1}\|^2 + \left(\frac{(1-\beta)^3|\gamma - 1|^2}{\beta}L^2 + 2(1-\beta)^2\delta_\gamma^2\right)\mathbb{E}\|x_t - x_{t-1}\|^2 + 2\beta^2\sigma^2. \tag{11}
$$

(When $\beta = 1$, $g_t = \nabla f_{\xi_t}(x_t)$ and $e_t = z_t$, so $\mathbb{E}\|e_t\|^2 \le \sigma^2$, and the subsequent steps still go through with $a = 0$.)

**Step 2) Descent inequality.** By $L$-smoothness of $f$, for any $x \in \mathbb{R}^d$ and $x^+ = x - \eta g$,

$$
\begin{aligned}
f(x^+) &\le f(x) + \langle \nabla f(x), x^+ - x\rangle + \frac{L}{2}\|x^+ - x\|^2 \\
&= f(x) - \eta\langle \nabla f(x), g\rangle + \frac{L}{2}\|x^+ - x\|^2 \\
&= f(x) - \frac{\eta}{2}\|\nabla f(x)\|^2 + \frac{\eta}{2}\|g - \nabla f(x)\|^2 + \left(\frac{L}{2} - \frac{1}{2\eta}\right)\|x^+ - x\|^2,
\end{aligned}
$$

where the last line uses the identity $-\eta\langle \nabla f(x), g\rangle = -\frac{\eta}{2}\|\nabla f(x)\|^2 - \frac{\eta}{2}\|g\|^2 + \frac{\eta}{2}\|g - \nabla f(x)\|^2$ and $\|x^+ - x\|^2 = \eta^2\|g\|^2$. Applying this with $x = x_t$, $x^+ = x_{t+1}$, and $g = g_t$ gives

$$
f(x_{t+1}) - f_{\inf} \le f(x_t) - f_{\inf} - \frac{\eta}{2}\|\nabla f(x_t)\|^2 + \frac{\eta}{2}\|e_t\|^2 + \left(\frac{L}{2} - \frac{1}{2\eta}\right)\|x_{t+1} - x_t\|^2. \tag{12}
$$

Define $H_t := f(x_t) - f_{\inf} + A\|e_t\|^2$ with $A > 0$. Then

$$
\begin{aligned}
\mathbb{E}[H_{t+1}] &= \mathbb{E}[f(x_{t+1}) - f_{\inf}] + A\mathbb{E}\|e_{t+1}\|^2 \\
&\overset{(12)}{\le} \mathbb{E}[f(x_t) - f_{\inf}] - \frac{\eta}{2}\mathbb{E}\|\nabla f(x_t)\|^2 + \frac{\eta}{2}\mathbb{E}\|e_t\|^2 + \left(\frac{L}{2} - \frac{1}{2\eta}\right)\mathbb{E}\|x_{t+1} - x_t\|^2 + A\mathbb{E}\|e_{t+1}\|^2 \\
&\overset{(11)}{\le} \mathbb{E}[f(x_t) - f_{\inf}] - \frac{\eta}{2}\mathbb{E}\|\nabla f(x_t)\|^2 + \left(\frac{\eta}{2} + A(1-\beta)\right)\mathbb{E}\|e_t\|^2 + 2\beta^2 A\sigma^2 \\
&\quad + \left(\frac{L}{2} - \frac{1}{2\eta} + \left(\frac{(1-\beta)^3|\gamma - 1|^2}{\beta}L^2 + 2(1-\beta)^2\delta_\gamma^2\right)A\right)\mathbb{E}\|x_{t+1} - x_t\|^2.
\end{aligned}
$$

Choose $A = \frac{\eta}{2\beta}$, so that $\frac{\eta}{2} + A(1-\beta) = A$ and $2\beta^2 A\sigma^2 = \beta\eta\sigma^2$. Thus,

$$\mathbb{E}[H_{t+1}] \leq \mathbb{E}[H_t] - \frac{\eta}{2}\mathbb{E}\|\nabla f(x_t)\|^2 + \beta\eta\sigma^2 + \left(\frac{L}{2} - \frac{1}{2\eta} + \left(\frac{(1-\beta)^3|\gamma-1|^2}{\beta}L^2 + 2(1-\beta)^2\delta_\gamma^2\right)\frac{\eta}{2\beta}\right)\mathbb{E}\|x_{t+1} - x_t\|^2.$$

**Step 3) Stepsize choices.**   It suffices to ensure

$$\frac{L}{2} - \frac{1}{2\eta} + \left(\frac{(1-\beta)^3|\gamma-1|^2}{\beta}L^2 + 2(1-\beta)^2\delta_\gamma^2\right)\frac{\eta}{2\beta} \leq 0,$$

which is equivalent to

$$\left(\frac{(1-\beta)^3|\gamma-1|^2}{\beta}L^2 + 2(1-\beta)^2\delta_\gamma^2\right)\frac{\eta^2}{\beta} + L\eta \leq 1.$$

By Lemma 3, this holds whenever $0 < \eta \leq \frac{1}{\sqrt{a}+L}$ with

$$a = \left(\frac{(1-\beta)^3|\gamma-1|^2}{\beta}L^2 + 2(1-\beta)^2\delta_\gamma^2\right)\frac{1}{\beta}.$$

Under this stepsize,

$$\mathbb{E}[H_{t+1}] \leq \mathbb{E}[H_t] - \frac{\eta}{2}\mathbb{E}\|\nabla f(x_t)\|^2 + \beta\eta\sigma^2.$$

$\square$

# D. Convergence Results of MARS

### D.1. Proof of Theorem 1

Let $\hat{x}_T$ be chosen uniformly at random from $\{x_0, x_1, \ldots, x_{T-1}\}$. Then

$$
\begin{aligned}
\mathbb{E}\|\nabla f(\hat{x}_T)\|^2 \quad &= \quad \frac{1}{T}\sum_{t=0}^{T-1}\mathbb{E}\|\nabla f(x_t)\|^2 \\
&\overset{\text{Lemma 4}}{\leq} \quad \frac{1}{T}\sum_{t=0}^{T-1}\left(\frac{2}{\eta}\left(\mathbb{E}[H_t] - \mathbb{E}[H_{t+1}]\right) + 2\beta\sigma^2\right) \\
&= \quad \frac{2}{\eta T}\sum_{t=0}^{T-1}\left(\mathbb{E}[H_t] - \mathbb{E}[H_{t+1}]\right) + 2\beta\sigma^2 \\
&= \quad \frac{2}{\eta T}\left(\mathbb{E}[H_0] - \mathbb{E}[H_T]\right) + 2\beta\sigma^2 \\
&\leq \quad \frac{2}{\eta T}\mathbb{E}[H_0] + 2\beta\sigma^2,
\end{aligned}
$$

since $H_T \geq 0$.

By definition,

$$
H_0 = f(x_0) - f_{\text{inf}} + \frac{\eta}{2\beta}\|g_0 - \nabla f(x_0)\|^2.
$$

If $g_0 = \frac{1}{B_{\text{init}}}\sum_{j=1}^{B_{\text{init}}}\nabla f_{\xi_j}(x_0)$, then by Assumption Assumption 2,

$$
\mathbb{E}\|g_0 - \nabla f(x_0)\|^2 \leq \frac{\sigma^2}{B_{\text{init}}}.
$$

Hence

$$
\mathbb{E}\|\nabla f(\hat{x}_T)\|^2 \quad \leq \quad \frac{2}{\eta T}\left(f(x_0) - f_{\text{inf}} + \frac{\eta}{2\beta}\cdot\frac{\sigma^2}{B_{\text{init}}}\right) + 2\beta\sigma^2.
$$

Choose $B_{\text{init}} = \left\lceil\frac{1}{\beta}\right\rceil$. Then $B_{\text{init}} \geq \frac{1}{\beta}$ and thus $\frac{1}{B_{\text{init}}} \leq \beta$, so

$$
\begin{aligned}
\mathbb{E}\|\nabla f(\hat{x}_T)\|^2 \quad &\leq \quad \frac{2}{\eta T}\left(f(x_0) - f_{\text{inf}} + \frac{\eta}{2}\sigma^2\right) + 2\beta\sigma^2 \\
&= \quad \frac{2}{\eta T}(f(x_0) - f_{\text{inf}}) + \frac{\sigma^2}{T} + 2\beta\sigma^2.
\end{aligned}
$$

Let $\Delta := f(x_0) - f_{\text{inf}}$. If we choose

$$
T = \frac{2\Delta}{\eta}\cdot\frac{1}{\epsilon^2} + \frac{\sigma^2}{\epsilon^2}, \qquad \beta = \min\left(1, \frac{\epsilon^2}{\sigma^2}\right),
$$

then $\frac{2\Delta}{\eta T} \leq \epsilon^2$, $\frac{\sigma^2}{T} \leq \epsilon^2$, and $2\beta\sigma^2 \leq 2\epsilon^2$, giving

$$
\mathbb{E}\|\nabla f(\hat{x}_T)\|^2 \leq 4\epsilon^2.
$$

**Gradient complexity.** Each iteration uses two stochastic gradients (at $x_t$ and $x_{t-1}$ with the same $\xi_t$), so the total number of gradient evaluations is

$$
B_{\text{init}} + 2T.
$$

With $\beta = \min\left(1, \frac{\epsilon^2}{\sigma^2}\right)$ we have

$$\frac{1}{\beta} = \max\left(1, \frac{\sigma^2}{\epsilon^2}\right),$$

hence $B_{\text{init}} = \left\lceil \frac{1}{\beta} \right\rceil = \mathcal{O}\left(\max\left(1, \frac{\sigma^2}{\epsilon^2}\right)\right)$, and

$$B_{\text{init}} + 2T = \mathcal{O}\left(\max\left(1, \frac{\sigma^2}{\epsilon^2}\right) + \frac{\Delta}{\eta\epsilon^2} + \frac{\sigma^2}{\epsilon^2}\right) = \mathcal{O}\left(1 + \frac{\sigma^2}{\epsilon^2} + \frac{\Delta}{\eta\epsilon^2}\right).$$

**Plugging in** $\eta = \frac{1}{L+\sqrt{a}}$. If $\eta = \frac{1}{L+\sqrt{a}}$, then $\frac{1}{\eta} = L + \sqrt{a}$ and

$$\sqrt{a} = \sqrt{\frac{(1-\beta)^3|\gamma-1|^2}{\beta^2}L^2 + \frac{2(1-\beta)^2}{\beta}\delta_\gamma^2} \leq \frac{(1-\beta)^{3/2}|\gamma-1|}{\beta}L + \frac{\sqrt{2}(1-\beta)}{\sqrt{\beta}}\delta_\gamma,$$

using $\sqrt{u+v} \leq \sqrt{u} + \sqrt{v}$. Therefore,

$$\frac{1}{\eta} \leq L + \frac{(1-\beta)^{3/2}|\gamma-1|}{\beta}L + \frac{\sqrt{2}(1-\beta)}{\sqrt{\beta}}\delta_\gamma.$$

Combining this with $\beta = \min\left(1, \frac{\epsilon^2}{\sigma^2}\right)$ yields the stated bound

$$B_{\text{init}} + 2T \leq \mathcal{O}\left(\frac{\sigma^2}{\epsilon^2} + \frac{L\Delta}{\epsilon^2} + \frac{\delta_\gamma\Delta\sigma}{\epsilon^3} + \frac{|\gamma-1|L\Delta\sigma^2}{\epsilon^4}\right).$$

### D.2. Comparison to Existing MVR Bounds

The gradient-evaluation complexity in Theorem 1 is stated as

$$B_{\text{init}} + 2T = \mathcal{O}\left(\frac{\sigma^2}{\epsilon^2} + \frac{L\Delta}{\epsilon^2} + \frac{\delta_\gamma\Delta\sigma}{\epsilon^3} + \frac{|\gamma-1|L\Delta\sigma^2}{\epsilon^4}\right).$$

In contrast, some references state the corresponding MVR bound under standard similarity in the form

$$\mathcal{O}\left(\frac{\sigma^2}{\epsilon^2} + \frac{(L+\delta)\Delta}{\epsilon^2} + \frac{\delta\Delta\sigma}{\epsilon^3}\right),$$

see, e.g., Fradin et al. (2026, Appendix I, Theorem I.1). The apparent discrepancy is only the extra $\delta\Delta/\epsilon^2$ term: it is *not* a fundamentally different rate, but rather a different way to upper bound the same stepsize-dependent expression.

Indeed, in our proof we use the stepsize choice $\eta = 1/(L+\sqrt{a})$ with

$$a = \left(\frac{(1-\beta)^3|\gamma-1|^2}{\beta}L^2 + 2(1-\beta)^2\delta_\gamma^2\right)\frac{1}{\beta},$$

hence

$$\frac{1}{\eta} = L + \sqrt{a} \leq L + \frac{(1-\beta)^{3/2}|\gamma-1|}{\beta}L + \frac{\sqrt{2}(1-\beta)}{\sqrt{\beta}}\delta_\gamma.$$

With $\beta = \min\{1, \epsilon^2/\sigma^2\}$ we have $\beta^{-1/2} = \max\{1, \sigma/\epsilon\}$, so a *uniform* (but looser) bound is obtained by using $(1-\beta) \leq 1$ and

$$\max\left\{1, \frac{\sigma}{\epsilon}\right\} \leq 1 + \frac{\sigma}{\epsilon},$$

which yields an additional contribution proportional to $\delta_\gamma$ in $1/\eta$, and hence an additional $\delta_\gamma\Delta/\epsilon^2$ term in the overall complexity. Concretely, one may equivalently state the following (slightly looser, but stylistically closer) bound:

$$B_{\text{init}} + 2T = \mathcal{O}\left(\frac{\sigma^2}{\epsilon^2} + \frac{(L+\delta_\gamma)\Delta}{\epsilon^2} + \frac{\delta_\gamma\Delta\sigma}{\epsilon^3} + \frac{|\gamma-1|L\Delta\sigma^2}{\epsilon^4}\right).$$

When $\gamma = 1$, $\delta_\gamma$ becomes the standard similarity constant $\delta$, and the above reduces (up to constants) to the form reported in Fradin et al. (2026, Appendix I, Theorem I.1).

Finally, note that in the variance-reduction regime $\epsilon \leq \sigma$ (equivalently $\beta = \epsilon^2/\sigma^2 < 1$), the extra $\delta_\gamma\Delta/\epsilon^2$ term is always dominated by $\delta_\gamma\Delta\sigma/\epsilon^3$ because $\sigma/\epsilon \geq 1$. This is why we omit $\delta_\gamma\Delta/\epsilon^2$ in the main displayed bound.

## D.3. A $T$-horizon Convergence Bound for MARS

**Theorem 2** (Generic $T$-horizon bound). *Consider MARS (Algorithm 1) for solving (1). Suppose Assumptions 1, 2 hold, and let $\delta_\gamma$ be the $\gamma$-similarity from Definition 1. Fix any $\beta \in (0,1]$ and $\gamma \in \mathbb{R}$, and choose a constant stepsize $\eta > 0$ satisfying*

$$0 < \eta \leq \frac{1}{L + \sqrt{a}}, \qquad a \triangleq \left( \frac{(1-\beta)^3 |\gamma - 1|^2}{\beta} L^2 + 2(1-\beta)^2 \delta_\gamma^2 \right) \frac{1}{\beta}.$$

*Let $\hat{x}_T$ be drawn uniformly at random from $\{x_0, \dots, x_{T-1}\}$. Then*

$$\mathbb{E}\|\nabla f(\hat{x}_T)\|^2 \leq \frac{2}{\eta T} \left( f(x_0) - f_{\inf} + \frac{\eta}{2\beta} \mathbb{E}\|g_0 - \nabla f(x_0)\|^2 \right) + 2\beta\sigma^2. \tag{13}$$

*In particular, if $g_0 = \frac{1}{B_{\text{init}}} \sum_{j=1}^{B_{\text{init}}} \nabla f_{\xi_j}(x_0)$ with $B_{\text{init}} \geq 1/\beta$, then $\mathbb{E}\|g_0 - \nabla f(x_0)\|^2 \leq \sigma^2/B_{\text{init}} \leq \beta\sigma^2$, and hence*

$$\mathbb{E}\|\nabla f(\hat{x}_T)\|^2 \leq \frac{2\Delta}{\eta T} + \frac{\sigma^2}{T} + 2\beta\sigma^2, \qquad \Delta \triangleq f(x_0) - f_{\inf}. \tag{14}$$

*Proof.* By Lemma 4, for $H_t = f(x_t) - f_{\inf} + \frac{\eta}{2\beta} \|e_t\|^2$ we have

$$\mathbb{E}[H_{t+1}] \leq \mathbb{E}[H_t] - \frac{\eta}{2} \mathbb{E}\|\nabla f(x_t)\|^2 + \beta\eta\sigma^2.$$

Summing from $t = 0$ to $T - 1$ and using telescoping yields

$$\frac{1}{T} \sum_{t=0}^{T-1} \mathbb{E}\|\nabla f(x_t)\|^2 \leq \frac{2}{\eta T} \left( \mathbb{E}[H_0] - \mathbb{E}[H_T] \right) + 2\beta\sigma^2 \leq \frac{2}{\eta T} \mathbb{E}[H_0] + 2\beta\sigma^2,$$

where we used $H_T \geq 0$. Finally, since $\hat{x}_T$ is uniform over $\{x_0, \dots, x_{T-1}\}$,

$$\mathbb{E}\|\nabla f(\hat{x}_T)\|^2 = \frac{1}{T} \sum_{t=0}^{T-1} \mathbb{E}\|\nabla f(x_t)\|^2,$$

which gives (13). The simplified bound (14) follows from $\mathbb{E}\|g_0 - \nabla f(x_0)\|^2 \leq \sigma^2/B_{\text{init}} \leq \beta\sigma^2$. $\square$

*Remark 1* (Why a random output iterate?). Lemma 4 directly gives a bound on the *average* $\frac{1}{T} \sum_{t=0}^{T-1} \mathbb{E}\|\nabla f(x_t)\|^2$. Returning $\hat{x}_T$ uniformly at random is a standard device to turn this average guarantee into a guarantee for a single iterate *without changing the bound*: $\mathbb{E}\|\nabla f(\hat{x}_T)\|^2 = \frac{1}{T} \sum_{t=0}^{T-1} \mathbb{E}\|\nabla f(x_t)\|^2$.

### D.3.1. PROOF OF COROLLARY 1

*Proof.* First note that since $(1 - \beta_T) \leq 1$, we have the crude upper bound

$$a = \left( \frac{(1-\beta_T)^3 |\gamma - 1|^2}{\beta_T} L^2 + 2(1-\beta_T)^2 \delta_\gamma^2 \right) \frac{1}{\beta_T} \leq \frac{|\gamma - 1|^2 L^2}{\beta_T^2} + \frac{2\delta_\gamma^2}{\beta_T}.$$

Hence $\sqrt{a} \leq \frac{|\gamma - 1|L}{\beta_T} + \frac{\sqrt{2}\delta_\gamma}{\sqrt{\beta_T}}$, so $\eta_T \leq \frac{1}{L+\sqrt{a}}$ and Theorem 2 applies. Using (14) gives

$$\mathbb{E}\|\nabla f(\hat{x}_T)\|^2 \leq \frac{2\Delta}{\eta_T T} + \frac{\sigma^2}{T} + 2\beta_T\sigma^2 = \frac{2\Delta}{T} \left( L + \frac{|\gamma - 1|L}{\beta_T} + \frac{\sqrt{2}\delta_\gamma}{\sqrt{\beta_T}} \right) + \frac{\sigma^2}{T} + 2\beta_T\sigma^2.$$

By the imposed horizon condition, $\beta_T = \beta_A + \beta_B \leq 1$, where

$$\beta_A = \sqrt{\frac{|\gamma - 1|L\Delta}{\sigma^2 T}}, \qquad \beta_B = 2^{-1/3} \left( \frac{\delta_\gamma \Delta}{\sigma^2 T} \right)^{2/3}.$$

Hence $\beta_T \geq \beta_A$ and $\beta_T \geq \beta_B$. Then

$$\frac{2\Delta|\gamma-1|L}{T\beta_T} + 2\sigma^2\beta_A \leq \frac{2\Delta|\gamma-1|L}{T\beta_A} + 2\sigma^2\beta_A = \frac{4\sigma\sqrt{|\gamma-1|L\Delta}}{\sqrt{T}},$$

and similarly

$$\frac{2\sqrt{2}\Delta\delta_\gamma}{T\sqrt{\beta_T}} + 2\sigma^2\beta_B \leq \frac{2\sqrt{2}\Delta\delta_\gamma}{T\sqrt{\beta_B}} + 2\sigma^2\beta_B = \frac{3\cdot 2^{2/3}(\delta_\gamma\Delta\sigma)^{2/3}}{T^{2/3}}.$$

Combining these estimates yields (9). $\qquad\square$

# E. Theoretical Advantage of MARS over MVR

## E.1. Why the quadratic surrogate is needed

A cruder consequence of Lemma 1 is

$$\delta_\gamma \leq \gamma\delta + (1-\gamma)L.$$

Using this bound in the non-common part of the displayed complexity bound gives

$$A\delta_\gamma + B(1-\gamma) \leq A\left[\gamma\delta + L\left(1 + \frac{\sigma}{\epsilon}\right)(1-\gamma)\right].$$

Therefore this crude certificate is no larger than the MVR term $A\delta$ only when

$$(1-\gamma)\left(L\left(1 + \frac{\sigma}{\epsilon}\right) - \delta\right) \leq 0.$$

Thus, unless $\delta \geq L(1 + \sigma/\epsilon)$, this linear bound certifies no improvement over $\gamma = 1$. This is why we use the sharper quadratic surrogate

$$J(\gamma) = A\sqrt{\gamma^2\delta^2 + (1-\gamma)^2 L^2} + B(1-\gamma),$$

which leads to Proposition 1.

## E.2. Proposition 1

**Proposition 1.** *Consider the objective function with respect to $\gamma \in [0,1]$:*

$$J(\gamma) = A\sqrt{\gamma^2\delta^2 + (1-\gamma)^2 L^2} + B(1-\gamma),$$

*where $A, B, \delta, L > 0$. Let $D := \delta^2 + L^2$ and $\gamma_\star = \text{argmin}_{\gamma\in[0,1]} J(\gamma)$. Then:*

- *If $B \geq A\delta$, then the minimum is at the boundary $\gamma_\star = 1$, and $J(\gamma_\star) = A\delta$.*

- *If $B < A\delta$, then the minimum is interior:*

$$\gamma_\star = \frac{L^2}{D} + \frac{BL\delta}{D\sqrt{A^2 D - B^2}},$$

  *and the optimal objective value satisfies*

$$J(\gamma_\star) = \frac{\delta\left(L\sqrt{A^2 D - B^2} + B\delta\right)}{D} \leq A\delta.$$

## E.3. Proof of Proposition 1

Let $D = \delta^2 + L^2$. We analyze the convexity and critical points of $J(\gamma)$.

**1. Convexity.** The function $f(\gamma) = \sqrt{\gamma^2\delta^2 + (1-\gamma)^2 L^2}$ is the Euclidean norm of the affine map $\gamma \mapsto (\gamma\delta, (1-\gamma)L)$. Since the composition of a convex function (the norm) and an affine map is convex, $f(\gamma)$ is convex. The term $B(1-\gamma)$ is linear and thus convex. Therefore, $J(\gamma)$ is convex on $\gamma \in [0,1]$.

**2. Boundary Solution ($\gamma_\star = 1$).** Since $J(\gamma)$ is convex, the minimum lies at the boundary $\gamma = 1$ if and only if the derivative at $\gamma = 1$ is non-positive ($J'(1) \leq 0$). Differentiating $J(\gamma)$:

$$J'(\gamma) = A\frac{2\gamma\delta^2 - 2(1-\gamma)L^2}{2\sqrt{\gamma^2\delta^2 + (1-\gamma)^2 L^2}} - B = A\frac{\gamma\delta^2 - (1-\gamma)L^2}{\sqrt{\gamma^2\delta^2 + (1-\gamma)^2 L^2}} - B.$$

Evaluating at $\gamma = 1$:

$$J'(1) = A\frac{1\cdot\delta^2 - 0}{\sqrt{\delta^2}} - B = A\delta - B.$$

Thus, the condition for $\gamma_\star = 1$ is $A\delta - B \leq 0 \iff B \geq A\delta$. In this case, $J(1) = A\delta$.

**3. Interior Solution ($\gamma_\star < 1$).** Assume $B < A\delta$. Since $J'(1) = A\delta - B > 0$, the minimizer cannot be $\gamma = 1$. Moreover, $J'(0) = -AL - B < 0$, so by convexity the minimizer is the unique point in $(0, 1)$ satisfying $J'(\gamma) = 0$.

Let $r(\gamma) = \sqrt{\gamma^2\delta^2 + (1-\gamma)^2 L^2}$, $D = \delta^2 + L^2$. The stationarity condition is

$$A\frac{\gamma\delta^2 - (1-\gamma)L^2}{r(\gamma)} = B.$$

Set $s = D\gamma - L^2 = \gamma\delta^2 - (1-\gamma)L^2$. One checks that $Dr(\gamma)^2 = s^2 + L^2\delta^2$. Thus the stationarity condition gives

$$A^2 s^2 = B^2 \frac{s^2 + L^2\delta^2}{D},$$

or equivalently $(A^2 D - B^2)s^2 = B^2 L^2 \delta^2$. Since $B > 0$ and the stationary point satisfies $s > 0$, we obtain $s = \frac{BL\delta}{\sqrt{A^2 D - B^2}}$. Therefore

$$\gamma_\star = \frac{L^2 + s}{D} = \frac{L^2}{D} + \frac{BL\delta}{D\sqrt{A^2 D - B^2}}.$$

The condition $B < A\delta$ implies $s < \delta^2$, and hence $\gamma_\star < 1$; clearly $\gamma_\star > 0$.

**4. Optimal Value and Inequality.** Substituting $\gamma_\star$ back into $J(\gamma)$ yields the expression:

$$J(\gamma_\star) = \frac{\delta\left(L\sqrt{A^2 D - B^2} + B\delta\right)}{D}.$$

To prove $J(\gamma_\star) \leq A\delta$, we rewrite the inequality as $L\sqrt{A^2 D - B^2} + B\delta \leq AD$. We apply the Cauchy-Schwarz inequality to vectors $\mathbf{u} = (L, \delta)$ and $\mathbf{v} = (\sqrt{A^2 D - B^2}, B)$:

$$\mathbf{u} \cdot \mathbf{v} \leq \|\mathbf{u}\|\|\mathbf{v}\| = \sqrt{L^2 + \delta^2}\sqrt{(A^2 D - B^2) + B^2} = \sqrt{D}\sqrt{A^2 D} = AD.$$

Dividing by $D$ and multiplying by $\delta$ recovers $J(\gamma_\star) \leq A\delta$.

# F. Auxiliary CIFAR-10 Probe

**Setup.** We use CIFAR-10 with a small CNN containing approximately $8.1 \times 10^5$ parameters. To keep the experiment close to the finite-sum stochastic optimization setting, we use the vanilla two-gradient $\gamma$-MVR update from Algorithm 1. We train for 20 epochs with batch size 128, learning rate 0.05, momentum coefficient 0.99 in the implementation, no weight decay, and a single random seed. For the checkpoint-level probe, we use a trajectory trained with $\gamma = 0.25$ and estimate local quantities every 500 steps using $M = 512$ sampled mini-batches.

**Local correction-scale estimate.** At each checkpoint, we compute the full-gradient difference

$$d_t = \nabla f(x_t) - \nabla f(x_{t-1})$$

and mini-batch differences

$$d_{B_m,t} = \nabla f_{B_m}(x_t) - \nabla f_{B_m}(x_{t-1}), \qquad m = 1, \ldots, M.$$

We estimate the local correction proxy

$$\widehat{h}_t(\gamma) = \frac{1}{M}\sum_{m=1}^{M} \|\gamma d_{B_m,t} - d_t\|^2,$$

and the corresponding local $\gamma$-similarity proxy

$$\widehat{\delta}^2_{\gamma,t} = \frac{\widehat{h}_t(\gamma)}{\|x_t - x_{t-1}\|^2}.$$

The local quantities

$$\widehat{L}^2_t = \frac{\|d_t\|^2}{\|x_t - x_{t-1}\|^2}, \qquad \widehat{\delta}^2_t = \frac{\frac{1}{M}\sum_m \|d_{B_m,t} - d_t\|^2}{\|x_t - x_{t-1}\|^2}$$

give the predicted minimizer

$$\widehat{\gamma}^\star_t = \frac{\widehat{L}^2_t}{\widehat{L}^2_t + \widehat{\delta}^2_t} = \frac{\|d_t\|^2}{\frac{1}{M}\sum_m \|d_{B_m,t}\|^2}.$$

*Table 1.* **Fixed-$\gamma$ CIFAR-10 sweep.** The best final training loss is attained at $\gamma = 0.25$. Test accuracy is reported for completeness, but this setup is not tuned as a CIFAR-10 generalization benchmark.

| $\gamma$ | Final train loss | Final test acc. | Best test acc. |
|---|---|---|---|
| 0 | 0.1925 | 72.90 | 75.23 |
| 0.025 | 0.1392 | 76.17 | 76.17 |
| 0.1 | 0.1023 | 74.25 | 74.68 |
| 0.25 | **0.0854** | 72.10 | 73.70 |
| 0.5 | 0.0911 | 70.37 | 71.79 |
| 0.75 | 0.1168 | 70.60 | 71.95 |
| 1 | 0.1415 | 70.69 | 71.72 |

**Fixed-$\gamma$ sweep.** As a complementary sanity check, we run a fixed-hyperparameter sweep over

$$\gamma \in \{0, 0.025, 0.1, 0.25, 0.5, 0.75, 1\}.$$

The final training loss is minimized at $\gamma = 0.25$ with final training loss $0.085$, while the MVR setting $\gamma = 1$ gives final training loss $0.142$ under the same hyperparameters. The best final test accuracy occurs at a different value, $\gamma = 0.025$, so we do not interpret this sweep as a tuned generalization benchmark. Its purpose is to show that the choice of $\gamma$ materially affects optimization in a theorem-aligned setting.

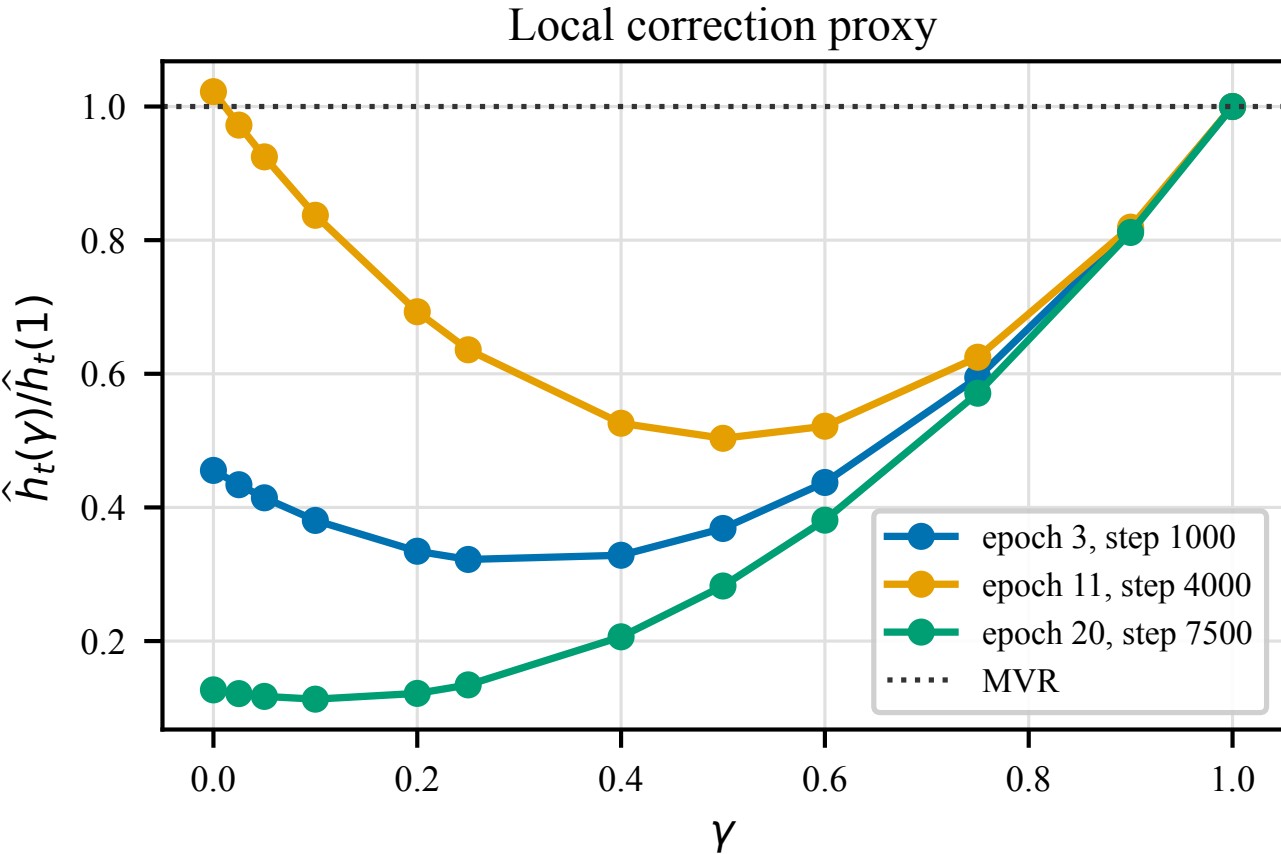

*Figure 4.* **Local correction proxy on CIFAR-10.** We plot $\widehat{h}_t(\gamma)/\widehat{h}_t(1)$ at representative checkpoints. Values below $1$ indicate that the scaled correction has smaller local discrepancy than the MVR correction $\gamma = 1$. This diagnostic is a checkpoint-level proxy for the numerator of $\gamma$-similarity and is not a global estimate of $\delta_\gamma$.

# G. LLM pretraining

**Motivation.** We study how performance varies with the scaling parameter $\gamma$ in a realistic LLM pretraining workload, since this is a primary modern use case. This experiment complements the theory by empirically characterizing sensitivity to $\gamma$ under a standard, well-tuned training protocol and by probing whether the preferred $\gamma$ appears to change across training stages, consistent with a regime-dependent tradeoff.

**Setup.** We train the 124M-parameter Llama-style model (Touvron et al., 2023) following the pretraining protocol of Semenov et al. (2025) and use their public training codebase.[1] We run the same training pipeline and hyperparameterization, except that we train on SlimPajama (Shen et al., 2023) instead of FineWeb (Penedo et al., 2024). Since the protocol is already carefully tuned, our goal is not exhaustive hyperparameter search but to observe the behavior of different $\gamma$ values under a sane and inexpensive setup; accordingly, we keep the remaining hyperparameters fixed rather than re-tuning for each $\gamma$. The training configuration is summarized in Table 2.

*Table 2.* **124M LLM pretraining configuration (fixed across the sweep).** We follow Semenov et al. (2025) and sweep only $\gamma$.

| Setting | Value |
|---|---|
| Model | Llama-style transformer, $\approx$ 124M params |
| Dataset | SlimPajama (Shen et al., 2023) |
| Sequence length | 512 |
| Batch | 256 sequences (length 512) |
| Token budget | $\approx$ 2.10B tokens |
| $\gamma$ values | $\{1, 0.1, 0.04, 0.025, 0.02, 0.01\}$ |
| Learning rates (AdamW; MARS) | $10^{-3}$; $3 \cdot 10^{-3}$ |
| Betas (AdamW; MARS) | $(0.8, 0.999)$; $(0.95, 0.99)$ |
| Weight decay | 0.1 |
| Warmup steps | 2000 |
| Scheduler | cosine |

**Results.** We report validation loss as a function of processed tokens in Figure 6.

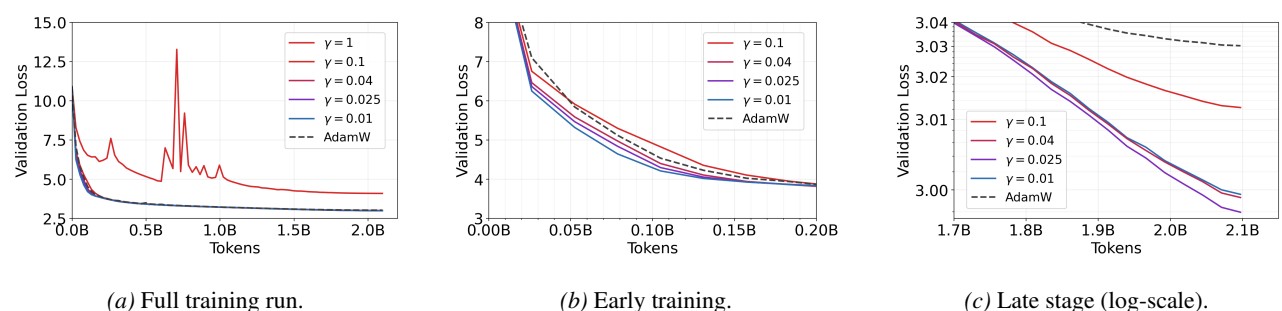

*(a)* Full training run.  *(b)* Early training.  *(c)* Late stage (log-scale).

*Figure 6.* **Validation loss vs. tokens for 124M pretraining under a $\gamma$-sweep.** We train a 124M Llama-style model on SlimPajama for $\approx$ 2.10B tokens (configuration in Table 2). Curves correspond to MARS-AdamW with different $\gamma$ values and an AdamW baseline.

Figure 6 shows three views of the same training run: Figure 6a is the full trajectory, Figure 6b zooms into the beginning, and Figure 6c zooms into the end. The goal is to compare token-efficiency and final loss across $\gamma$ values. Overall, the validation-loss trajectory appears sensitive to $\gamma$, with several $\gamma < 1$ settings tracking below AdamW for substantial portions of training.

Figure 6b highlights the early stage: in our runs, smaller $\gamma$ values tend to reduce validation loss more quickly as a function of tokens processed in this regime.

Figure 6c highlights the end of training. Here, differences become subtler and the ordering among $\gamma$ values can shift relative to the early stage (e.g., a $\gamma$ that looks best early need not be best late). This suggests that as training progresses the iterates may traverse regions with different local properties, which can change which $\gamma$ best balances the relevant terms in our bound (e.g., the $\delta_\gamma$-dependent term versus the $(1 - \gamma)$-dependent penalty), though the differences in final loss among small $\gamma$ values

---

[1] https://github.com/epfml/llm-optimizer-benchmark

are relatively small in this run. Finally, Figure 6a shows that the $\gamma = 1$ run exhibits pronounced instability under the fixed hyperparameters, indicating that the out-of-the-box $\gamma = 1$ setting may require re-tuning.

To summarize the main observations more directly: (i) small $\gamma$ values can outperform AdamW, which is visible near the end of training in Figure 6c, where several $\gamma < 1$ curves lie below AdamW; (ii) different small $\gamma$ values behave differently and there is an optimal choice, and the fact that the early-stage ordering in Figure 6b differs from the late-stage ordering in Figure 6c is consistent (though not conclusive) with stage-dependent local properties affecting which terms dominate in the bound; and (iii) the classic MVR setting ($\gamma = 1$) can be sensitive to hyperparameters and exhibit instability without dedicated tuning, as illustrated by the full trajectory in Figure 6a.

**Implications.**

- Small $\gamma$ values can outperform AdamW and the classic MVR setting ($\gamma = 1$) in this workload, indicating that the choice of $\gamma$ is practically important.

- Different small $\gamma$ values behave differently and there is an optimal choice; the early/late comparison in Figure 6b and Figure 6c suggests (cautiously) that changes in local properties along training can affect which $\gamma$ is preferred, even though the final-loss differences among small $\gamma$ values are relatively small here.

- The classic MVR setting ($\gamma = 1$) is unstable under the fixed hyperparameters (Figure 6a), suggesting that $\gamma = 1$ can be hyperparameter-sensitive and may require dedicated tuning.

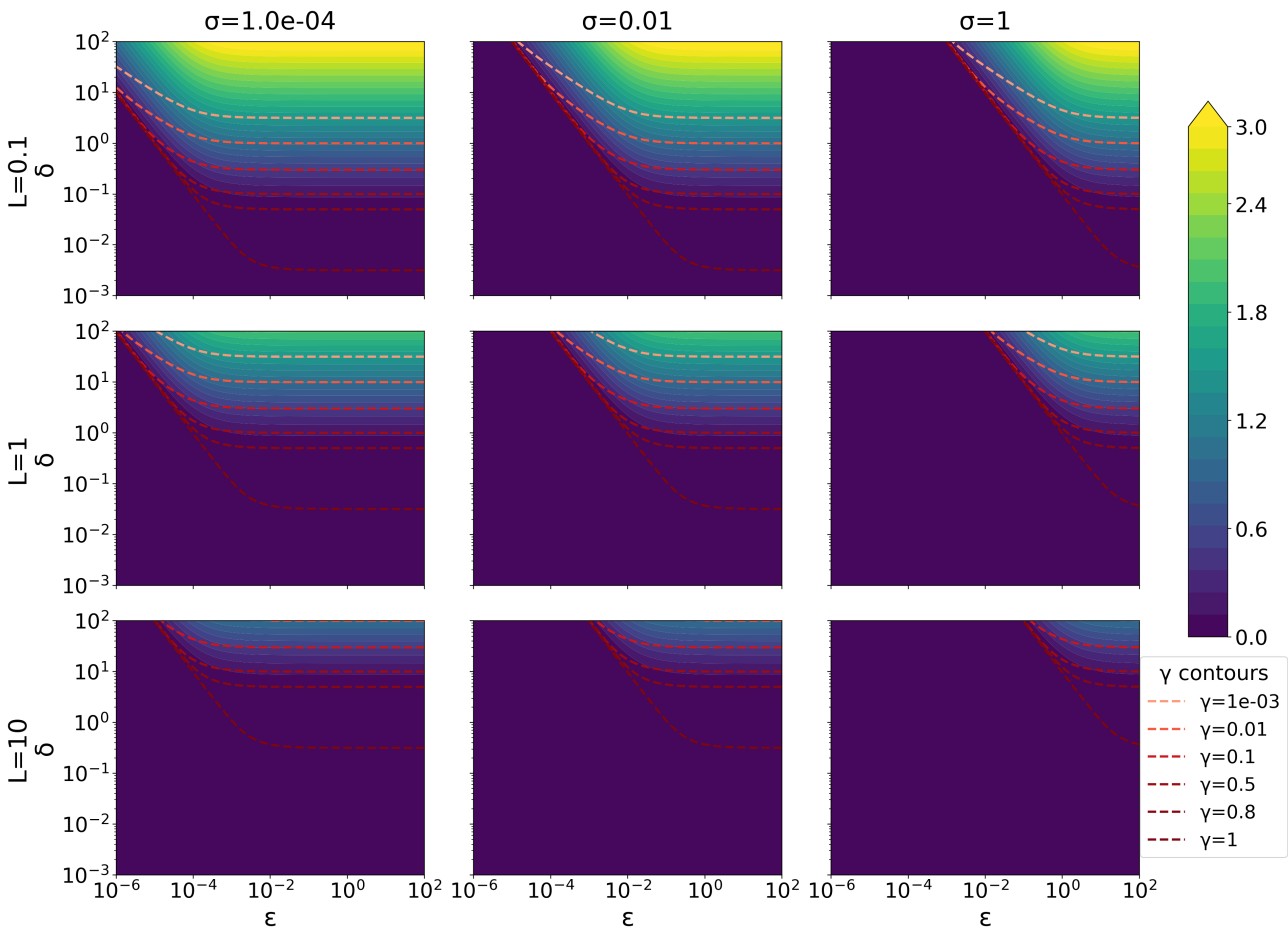

*Figure 7.* **Bound-implied** MARS **vs.** MVR **speedup (full grid).** Rows correspond to $L \in \{0.1, 1, 10\}$, columns correspond to $\sigma \in \{10^{-4}, 10^{-2}, 1\}$. Each panel plots $\log_{10}\big((A\delta)/J(\gamma_\star)\big)$ on a log–log grid over $\epsilon$ (horizontal) and $\delta$ (vertical), with $J(\gamma) = A\sqrt{\gamma^2\delta^2 + (1-\gamma)^2 L^2} + B(1-\gamma)$, $A = \sigma/\epsilon^3$, $B = L\sigma^2/\epsilon^4$ (here $\Delta = 1$). Dashed curves show $\gamma_\star$ contour levels.

## H. Theory visualization: bound-implied MARS vs. MVR speedup

**Motivation.** The analysis in the main text shows that, in a low-target-accuracy regime, MARS can attain a lower gradient/complexity bound than MVR due to the way $\gamma$ trades off a $\delta_\gamma$-dependent term against a $(1-\gamma)$-dependent penalty. This appendix figure makes that tradeoff concrete by visualizing, across a wide range of problem/accuracy parameters, (i) how much the bound improves when optimizing $\gamma$, and (ii) which $\gamma_\star$ the surrogate prefers.

**Setup (surrogate objective and normalization).** For fixed $(L, \sigma, \epsilon, \delta)$ we consider the one-dimensional convex objective

$$J(\gamma) \;=\; A\sqrt{\gamma^2\delta^2 + (1-\gamma)^2 L^2} \;+\; B(1-\gamma), \qquad \gamma \in [0, 1],$$

with $\Delta = 1$ and the identifications

$$A = \frac{\sigma}{\epsilon^3}, \qquad B = \frac{L\sigma^2}{\epsilon^4}.$$

The MVR baseline corresponds to $\gamma = 1$, in which case $J(1) = A\delta$. We report the *bound-implied speedup* of optimizing $\gamma$ relative to this baseline,

$$\log_{10} \text{speedup} \;=\; \log_{10}\left(\frac{A\delta}{J(\gamma_\star)}\right),$$

so that $\log_{10}$ speedup $= 0$ means "no improvement over $\gamma = 1$" and larger values indicate stronger improvement in the surrogate.

**What is shown.**    Figure 7 is a $3 \times 3$ grid of $\log_{10}$ speedup heatmaps. Rows correspond to $L \in \{0.1, 1, 10\}$, columns correspond to $\sigma \in \{10^{-4}, 10^{-2}, 1\}$. Each panel is evaluated on a log–log grid over $\epsilon$ (horizontal axis) and $\delta$ (vertical axis). Dashed curves overlay contour lines of the optimizer $\gamma_\star$ (selected levels shown in the legend); smaller $\gamma_\star$ indicates that the surrogate prefers a more aggressive deviation from the MVR baseline.

**Qualitative trends and interpretation.**    The grid exhibits three robust qualitative behaviors:

- **Larger $\epsilon$ and $\delta$ yield larger speedups.** Across all panels, the heatmaps brighten toward the upper-right: increasing $\epsilon$ and $\delta$ tends to decrease the relative importance of the $(1 - \gamma)$-penalty term (through $B$) compared to the baseline $A\delta$, making $\gamma < 1$ more beneficial in the surrogate.

- **A diagonal transition is governed by the ratio $L\sigma/(\epsilon\delta)$.** The boundary between the dark region ($\log_{10}$ speedup $\approx 0$, $\gamma_\star \approx 1$) and the brighter improvement region aligns approximately along diagonals in $(\epsilon, \delta)$. This is consistent with the surrogate depending on

$$\frac{B}{A\delta} = \frac{L\sigma}{\epsilon\delta},$$

so that improvement emerges when $\epsilon\delta$ is sufficiently large relative to $L\sigma$. The $\gamma_\star$ contours track this transition: $\gamma_\star$ drops below 1 as one moves into the improvement regime.

- **Dependence on $L$ and $\sigma$: larger $L$ or $\sigma$ shifts the improvement regime "outward".** Increasing either $L$ (down the rows) or $\sigma$ (across columns) increases $L\sigma$ and hence increases $B/(A\delta)$ at fixed $(\epsilon, \delta)$. In the plots, this manifests as a shift of the improvement region toward larger $\epsilon$ and/or larger $\delta$, and a corresponding tendency for $\gamma_\star$ to remain closer to 1 unless $\epsilon\delta$ is sufficiently large.

**Takeaway.**    Overall, the figure supports the qualitative message that the surrogate favors smaller $\gamma$ (and predicts larger improvements over MVR) in regimes with larger $\epsilon$ and $\delta$, while larger $L$ and $\sigma$ make improvement harder to realize unless $\epsilon\delta$ grows commensurately. In the main text we include a compact $1 \times 2$ view at fixed $L = 0.1$ and two representative $\sigma$ values; this appendix provides the full sweep.

