# OpenReview forum: "Understanding MARS: When Scaling Momentum Provably Helps"
_ICML.cc/2026/Conference — ICML 2026 regular_

### Official Review · Reviewer_N3hV · 2026-03-10

**Soundness:** 2
**Presentation:** 3
**Significance:** 3
**Originality:** 3
**Overall Recommendation:** 4
**Confidence:** 3

**Summary:**

The paper addresses a theoretical gap in understanding why the MARS optimizer consistently outperforms standard Momentum-based Variance Reduction (MVR) and AdamW in large language model (LLM) training. Existing theories fail to explain the benefits of MARS's multiplicative coefficient $\gamma$, which scales the momentum correction term, because standard similarity assumptions penalize any $\gamma < 1$. To resolve this, the authors introduce a novel theoretical framework centered on the $\gamma$-similarity condition. This refined condition generalizes both standard similarity and smoothness, proving that with an appropriately tuned $\gamma \in [0, 1]$, MARS achieves strictly lower gradient complexity than MVR. The theoretical findings are corroborated by empirical experiments on a 124M parameter GPT-style model, demonstrating that optimizing $\gamma$ yields substantial gains in token efficiency.

**Compliance With Llm Reviewing Policy:**

Affirmed.

**Final Justification:**

I am inclined to raise my score because the additional auxiliary CNN experiment on CIFAR-10 provided in the rebuttal offers useful empirical support for the paper’s theoretical claims and partially bridges the gap between theory and experiment. I would recommend that the authors incorporate this result into the revised version and clarify the acknowledged limitations and scope more explicitly.

**Key Questions For Authors:**

See the detailed weaknesses above, particularly regarding the empirical validation of the $\gamma$-similarity assumption and the tuning of baselines.

**Limitations:**

The authors should include a dedicated Limitations section or append one to the conclusion.

**Strengths And Weaknesses:**

>**Disclosure (Policy B):** In compliance with the ICML 2026 LLM Policy B (Permissive), to improve linguistic accuracy and communication efficiency, I used a LLM to translate and polish this review. All opinions, technical analyses, and judgments were written by me in my native language; the LLM was used only for translation and language refinement.

**Strengths**

Bridging the gap between the empirical success of MARS and its theoretical justification is a highly relevant problem in modern optimization. The theoretical foundation is robust and successfully diagnoses why previous analyses failed to capture MARS's empirical benefits. By shifting the perspective from treating $(\gamma-1)d_\xi$ as a penalty to directly analyzing the aggregate term $(\gamma d_\xi - d)$, the authors prove lower gradient complexity. The convergence theorem cleanly recovers standard $\mathcal{O}(1/\epsilon^3)$ complexity for MVR ($\gamma=1$) and $\mathcal{O}(1/\epsilon^4)$ for stochastic momentum ($\gamma=0$), confirming theoretical consistency.

**Weaknesses**

- While the $\gamma$-similarity assumption is elegant, it is not empirically validated. The paper does not provide any measurement or estimation of $\delta_\gamma$ for the actual LLM training tasks considered in the experiments. Without such estimates, it is unclear whether the theoretical optimal $\gamma_{\star}$ from Lemma 2 matches the empirically optimal $\gamma$ observed in the GPT experiments. This disconnect between theory and experiment weakens the overall narrative.

- The experiments lack comprehensive baseline tuning. The authors state that classic MVR ($\gamma=1$) can be unstable without dedicated tuning, yet they fix hyperparameters based on prior work and solely sweep $\gamma$. This raises the possibility that smaller $\gamma$ values simply act as a regularization mechanism compensating for suboptimal base hyperparameters.

- While the theory is significant, the modest scale of the empirical validation restricts the immediate impact on the LLM practitioner community. Demonstrating these gains at the 1B+ parameter scale is generally expected for papers making strong claims about LLM pretraining efficiency.

- There are several noticeable inconsistencies and editorial issues throughout the manuscript. For example, the main text contains repeated or broken sentences (e.g., lines 202 and 206). Furthermore, the initial definition of $h(x, y ; \gamma):=E_{\xi}\left\|d_{\xi}(x, y)-d(x, y)\right\|^{2}$ is inconsistent with the expression actually used later in Lemma 1, which introduces the scaling factor $\gamma$: $h(x, y ; \gamma):=E_{\xi}\left\|\gamma d_{\xi}(x, y)-d(x, y)\right\|^{2}$. These issues detract from the paper's polish and readability.

---

> ### Author Rebuttal · Authors · 2026-03-31
>
> Thank you for the careful review. We appreciate that you found the core theoretical contribution meaningful. We also agree that the two most important issues are:
> (i) tightening the theory/experiment connection, and
> (ii) clarifying the scope of the empirical comparison.
>
> > The similarity assumption is elegant, but it is not empirically validated; no estimate of $\delta_{\gamma}$ no check whether predicted $\gamma^\star$ matches the empirical one.
>
> We agree this was the most important gap in the submission, and we have used the rebuttal period to address it with a new theorem-aligned auxiliary experiment.
>
> On a simple CNN on CIFAR-10 task where full gradients are computable, we estimate the quantities in Lemma 2 from stochastic gradient-difference statistics collected along training. At selected checkpoints, we compare consecutive end-of-epoch iterates $x_t,x_{t-1}$, compute
> $d=\nabla f(x_t)-\nabla f(x_{t-1})$,
> and repeatedly evaluate same-batch stochastic differences
> $d_\xi=\nabla f_\xi(x_t)-\nabla f_\xi(x_{t-1})$
> on many mini-batches. This gives empirical estimates of
> $L \approx \|d\|/\|x_t-x_{t-1}\|$
> and
> $\delta^2 \approx \mathbb{E}\|d_\xi-d\|^2/\|x_t-x_{t-1}\|^2$.
> Using worst-case checkpoint values and Lemma 2,
> $\gamma^\star=L^2/(\delta^2+L^2)$,
> we obtain a practical prediction of roughly **$\gamma^\star \in [0.5,0.6]$**. In the same setup, this is close to the empirically best fixed $\gamma$ over our tested grid (**$\gamma=0.5$**). The figure is here: https://ibb.co.com/bjFTrcCR
>
> We agree this is still not a direct estimate for the 124M LLM run; our claim is narrower and more realistic. The new experiment validates the theory in the **same algorithmic setting as the theorem**, and shows that the theory-derived $\gamma^\star$ is not merely formal.
>
> > The experiments lack comprehensive baseline tuning; smaller $\gamma$ might simply compensate for suboptimal $\gamma=1$ hyperparameters.
>
> This is a fair concern, and we will clarify the scope more carefully.
>
> Our 124M GPT experiment was designed as an **isolation study**: we adopt the public pretraining protocol and already-tuned non-$\gamma$ hyperparameters of Semenov et al. (2025), and then sweep only $\gamma$ inside the same optimizer stack. This isolates the effect of the MARS scaling coefficient, but we agree it does **not** constitute a fully re-tuned leaderboard comparison for $\gamma=1$.
>
> We will therefore temper the wording around instability and make the intended claim explicit: the experiment is meant to show that the practical phenomenon motivating the theory is real, not to claim that we have exhaustively optimized every baseline. At the same time, our theoretical message is stronger than a generic “regularization effect”: Proposition 1 predicts that $\gamma<1$ is **not uniformly better** than $\gamma=1$; depending on the regime, the optimum can move back to $\gamma=1$. This regime dependence is precisely what we observe empirically.
>
> > The empirical scale is modest for LLM claims.
>
> We agree that the 124M study is supportive rather than exhaustive. The intent of this submission is to provide the first theoretical explanation of why $\gamma<1$ can be provably advantageous at all. For broader practical benchmarking, MARS has already been evaluated in prior work, and recent benchmark studies include MARS at scales up to **1.2B** parameters, see Wen et al. (2025). We will make this distinction clearer.
>
> > There are editorial issues / broken sentences / inconsistent definition of $h(x,y;\gamma)$.
>
> Thank you for catching these. We agree that these issues hurt the presentation. In the revision/camera-ready we will:
> 1. fix the typo in the definition of $h(x,y;\gamma)$ so it consistently reads $\mathbb{E}\|\gamma d_\xi(x,y)-d(x,y)\|^2$,
> 2. rewrite the broken/repeated sentences around Lines 202–206,
> 3. add a dedicated Limitations paragraph/section.
>
> Overall, we agree with your diagnosis: the theory is strong, and the main needed improvement is a cleaner bridge to practice. The new checkpoint-level empirical validation is intended precisely to close that gap.

---

> > ### Author Rebuttal · Reviewer_N3hV · 2026-04-02
> >
> > Thank you for the detailed answer. I'll increase my score.

---

### Official Review · Reviewer_mXHC · 2026-03-13

**Soundness:** 4
**Presentation:** 3
**Significance:** 3
**Originality:** 3
**Overall Recommendation:** 4
**Confidence:** 3

**Summary:**

This paper studies the theoretical properties of the MARS optimizer, a recently proposed optimization method that empirically improves over MVR and AdamW in large-scale training tasks such as large language model (LLM) pretraining. The key idea of MARS is to introduce a scaling coefficient γ that multiplies the stochastic gradient difference term in the momentum-based variance reduction update.

The authors proceed to explore a central concept: explaining why this seemingly simple scaling mechanism improves convergence over the original Momentum-based Variance Reduction (MVR) method. The manuscript addresses the concept by introducing a refined theoretical condition called γ-similarity, which generalizes the standard similarity assumption commonly used in stochastic optimization analyses. Under this new assumption, the authors derive nonconvex convergence guarantees showing that, for appropriate choices of γ, the gradient complexity of MARS is strictly smaller than that of MVR.

The theoretical analysis provides explicit gradient complexity bounds for MARS and shows that the algorithm interpolates between stochastic momentum methods and MVR depending on the value of γ. In addition to the theoretical results, the paper presents empirical experiments in GPT-style language model pretraining, showing that properly tuned γ values improve token efficiency compared with both MVR and AdamW baselines.

Overall, the work attempts to bridge a gap between empirical observations about the effectiveness of the MARS optimizer and the lack of a theoretical explanation for its improved convergence behavior.

**Compliance With Llm Reviewing Policy:**

Affirmed.

**Key Questions For Authors:**

1. Practical estimation of γ-similarity constants.
How can the γ-similarity constant δγ be estimated or approximated in real deep learning training scenarios?

2. Scalability of empirical evaluation.
Have the authors considered evaluating the method on larger LLM models (e.g., billions of parameters) to validate the empirical advantages at modern scales?

3. Sensitivity to γ selection.
How sensitive is training performance to different γ values in practice? Are there heuristics or adaptive strategies for selecting γ during training?

**Limitations:**

Yes. The authors include an impact statement describing how improved optimization techniques may lead to more efficient training of large-scale models, while not introducing significant direct societal risks.

**Strengths And Weaknesses:**

**Strengths**

1. Clear motivation and theoretical gap identification

The paper identifies an interesting discrepancy between theory and practice: although MARS empirically improves over MVR in large-scale training, existing theoretical analyses do not explain this advantage. The paper provides a well-motivated attempt to close this gap.

2. Introduction of the γ-similarity framework

The main technical contribution is the γ-similarity condition, which generalizes both smoothness and standard similarity assumptions. This new condition provides a more refined way of analyzing scaled gradient-difference estimators and allows the authors to derive stronger theoretical guarantees for MARS.

3. Improved convergence analysis

Using the γ-similarity condition, the authors derive explicit gradient complexity bounds and show that for appropriate γ values the complexity of MARS can be strictly smaller than that of MVR. This provides the first theoretical explanation for MARS's empirical advantages.

4. Insightful theoretical interpretation of γ

The analysis demonstrates how the scaling parameter γ controls the trade-off between variance reduction and bias in gradient estimators. The theoretical results help clarify when smaller γ values can lead to improved optimization behavior.

5. Empirical validation on LLM pretraining

The experiments provide evidence that smaller γ values can outperform AdamW and MVR in language model pretraining. The γ sweep experiments illustrate the existence of an optimal γ value for token efficiency during training.

**Weaknesses**

1. Theoretical assumptions may be difficult to verify in practice

Although γ-similarity is theoretically elegant, it is not clear how easily the associated constants (e.g., δγ) can be estimated or validated for real deep learning models. The practical implications of the assumption therefore remain somewhat unclear.

2. Limited empirical evaluation

The empirical section focuses on a relatively small-scale GPT-style model (124M parameters). While this provides useful insights, additional experiments on larger models or more diverse workloads would strengthen the empirical claims.

3. Incremental empirical contribution

The main novelty of the paper is theoretical. The experimental results are supportive but relatively limited compared to the theoretical development.

4. Dense theoretical exposition

Some sections of the theoretical analysis are quite dense and difficult to follow. Additional intuition or intermediate explanations would improve readability for a broader machine learning audience.

5. Sensitivity to hyperparameter selection

While the theory explains the benefits of properly tuned γ values, practical guidelines for selecting γ remain somewhat limited.

---

> ### Author Rebuttal · Authors · 2026-03-31
>
> Thank you for the careful and balanced review. We are glad that you found the theoretical gap well-motivated and the $\gamma$-similarity framework useful for understanding when scaling the momentum correction can help.
>
> > How can the $\gamma$-similarity constant be estimated or approximated in practice?
>
> We agree that this was underdeveloped in the submission, and we now have a concrete preliminary answer.
>
> While the exact global constants in the theorem are not directly accessible during real large-scale training, one can estimate **trajectory-level proxies** from gradient-difference statistics. In a new auxiliary CNN on CIFAR-10 experiment, at selected checkpoints we compute the full gradient difference
> $d=\nabla f(x_t)-\nabla f(x_{t-1})$
> and repeated same-batch stochastic differences
> $d_\xi=\nabla f_\xi(x_t)-\nabla f_\xi(x_{t-1})$.
> From these we estimate
> $L \approx \|d\|/\|x_t-x_{t-1}\|$
> and
> $\delta^2 \approx \mathbb{E}\|d_\xi-d\|^2/\|x_t-x_{t-1}\|^2$,
> and then plug them into Lemma 2:
> $\gamma^\star=L^2/(\delta^2+L^2)$.
>
> Using worst-case values over the measured checkpoints gives a practical run-level prediction of about **$\gamma^\star \in [0.5,0.6]$**. In the same setup, this is close to the empirically best fixed $\gamma$ over our grid (**$\gamma=0.5$**). We also observe that the local preferred $\gamma$ evolves during training rather than staying constant. The figure is here: https://ibb.co.com/bjFTrcCR
>
> For LLM training, the same idea would require replacing the exact full gradient with a large-batch or accumulated-gradient approximation. We do not claim to have solved that practical design problem in this paper; rather, our contribution is to establish the theoretical foundation that such a method should target.
>
> > The empirical section is limited to a 124M GPT-style model.
>
> We agree that the empirical section is intentionally narrower than the theory. Our goal with the 124M experiment was not to present a new large-scale optimizer benchmark, but to provide a controlled practical check that the fixed-$\gamma$ phenomenon predicted by the theory is visible in a realistic LLM training pipeline.
>
> For larger-scale practical benchmarking, MARS itself has already been studied extensively in prior work, and more recent benchmark work evaluates MARS up to **1.2B parameters**, see Wen et al. (2025). We will make this distinction clearer: the main novelty of this paper is the **theory**, while the experiment is meant as supporting evidence rather than an exhaustive empirical campaign.
>
> > How sensitive is performance to $\gamma$? Are there heuristics or adaptive strategies?
>
> Our current view is that the preferred $\gamma$ is **stage-dependent/local**, rather than universal. Figure 2 already suggests this because the early-stage and late-stage orderings differ. Proposition 1 gives the corresponding theoretical intuition: smaller $\gamma$ is favored when the $\delta_\gamma$-driven gain dominates the $(1-\gamma)$-dependent penalty, whereas harder regimes push the optimum back toward $\gamma=1$.
>
> The new auxiliary experiment strengthens this point: the empirically estimated preferred $\gamma$ changes during training rather than staying fixed. We therefore believe the right practical heuristic is **not** a single universal default, but either (i) selecting $\gamma$ from local gradient-difference statistics, or (ii) developing an adaptive schedule. We view this as an important future direction built directly on the present theory.
>
> > Some parts of the theory are dense.
>
> We agree. In the revision/camera-ready, we will improve accessibility by adding:
> 1. a short intuition paragraph before Lemma 1 explaining why standard similarity is too coarse for MARS,
> 2. a proof roadmap before the main theorem, and
> 3. a clearer plain-language explanation of the $\delta_\gamma$ vs. $(1-\gamma)$ trade-off.
>
> We will also add a dedicated limitations paragraph stating explicitly that the theorem is for vanilla two-gradient $\gamma$-MVR, that practical estimation relies on local proxies rather than exact global constants, and that the empirical study is supportive rather than comprehensive.

---

> > ### Author Rebuttal · Reviewer_mXHC · 2026-04-03
> >
> > The rebuttal solved my concerns, so I maintain my score.

---

### Official Review · Reviewer_EpCR · 2026-03-14

**Soundness:** 3
**Presentation:** 3
**Significance:** 2
**Originality:** 3
**Overall Recommendation:** 4
**Confidence:** 3

**Summary:**

MARS scales the momentum correction term in MVR by a coefficient $\gamma \in [0,1]$ and empirically outperforms both MVR and AdamW in LLM training, but existing theory (Yuan et al., 2025) cannot explain this — under standard assumptions, MARS with $\gamma < 1$ is provably worse than MVR. This paper introduces a refined assumption called $\gamma$-similarity that captures the interaction between the scaling coefficient and gradient noise structure. Under this condition, the authors derive convergence bounds showing MARS with an appropriately chosen $\gamma$ achieves strictly lower gradient complexity than MVR, and characterize the optimal $\gamma$ in closed form. Experiments on 124M GPT pretraining corroborate that an optimal $\gamma < 1$ exists.

**Compliance With Llm Reviewing Policy:**

Affirmed.

**Final Justification:**

The paper is technically sound. I acknowledge their theoretical analysis and insightful interpretation of $\gamma$ while the practical impacts remain modest.

**Key Questions For Authors:**

1) Can you estimate $\delta_\gamma$ empirically during training and check whether the predicted $\gamma_\star$ matches the empirically optimal $\gamma$?
2) Does the analysis extend to MARS-AdamW? Does $\gamma$-similarity hold in the preconditioned space?
3) The early-stage and late-stage optimal $\gamma$ orderings differ (Figure 2). Does the theory predict this?

**Limitations:**

yes

**Strengths And Weaknesses:**

Strengths:
1) Addresses a genuine gap over Yuan et al. (2025): their bounds give $O(1/\epsilon^3)$ for both MARS and MVR (no separation), and their adaptive $\gamma_t$ requires infeasible full-gradient quantities. This paper separates the two with a fixed, practical $\gamma$ under $\gamma$-similarity.

2) The paper is well-written with a clear progression from gap identification to new assumption to convergence results. Appendix C.2 carefully reconciles the bounds with prior MVR results.


Weaknesses:
1) The practical value of this paper seems unclear to me. The optimal $\gamma_\star$ requires $\delta$ and $L$ which are inaccessible, so practitioners still just grid-search $\gamma$. The analysis covers vanilla $\gamma$-MVR, not MARS-AdamW. The $\gamma$-similarity condition is specific to MARS and does not appear to generalize to other momentum scaling schemes. For the contribution to have broader impact, the analysis would need to offer insights that transfer beyond MARS, e.g., general principles for when momentum scaling helps, but the paper stays within the scope of this one algorithm.

2) Theory and experiments are disconnected. Both suggest an optimal $\gamma$ exists, but the paper never links them: no empirical estimate of $\delta_\gamma$, no check of predicted vs actual $\gamma_\star$, no verification that $\gamma$-similarity holds tighter than the Lemma 2 upper bound. The theory "explains" the experiments only in the loosest sense.

3) The $\gamma$-similarity condition essentially assumes that the scaled correction is a better estimator, then shows MARS converges faster as a consequence. This leaves the more interesting question unanswered: what properties of a problem (e.g., curvature, noise structure) make $\delta_\gamma < \delta$? The paper only answers this for quadratics (Example 1).

---

> ### Author Rebuttal · Authors · 2026-03-31
>
> Thank you for the careful review. We are especially encouraged that you view the central gap over prior MARS theory as genuine and the $\gamma$-similarity framework as technically meaningful. We agree that the main place to strengthen the paper is the **theory/experiment bridge**, and we have used the rebuttal period to address exactly that.
>
> > Can you estimate $\delta_\gamma$ empirically during training and check whether the predicted $\gamma^\star$ matches the empirically optimal one?
>
> Yes, at least in a theorem-aligned small-scale setting, and we now have preliminary evidence in exactly this direction.
>
> Because the theorem is for vanilla two-gradient $\gamma$-MVR, we ran an auxiliary experiment on a simple CNN trained on CIFAR-10 in a finite-sum regime where full gradients are computable. At selected checkpoints, we take two consecutive end-of-epoch iterates $x_t,x_{t-1}$, compute the full gradient difference
> $d=\nabla f(x_t)-\nabla f(x_{t-1})$,
> and repeatedly compute same-batch stochastic differences
> $d_\xi=\nabla f_\xi(x_t)-\nabla f_\xi(x_{t-1})$
> over many mini-batches. This lets us estimate
> $L \approx \|d\|/\|x_t-x_{t-1}\|$
> and
> $\delta^2 \approx \mathbb{E}\|d_\xi-d\|^2/\|x_t-x_{t-1}\|^2$.
> We then plug worst-case checkpoint estimates into Lemma 2,
> $\gamma^\star=L^2/(\delta^2+L^2)$.
>
> In these 30-epoch runs, the resulting theory-predicted $\gamma^\star$ falls in the range **$0.5$--$0.6$**, and this is close to the empirically best fixed $\gamma$ in the same setup (**$\gamma=0.5$** over our tested grid). In addition, when we plot the checkpoint-level quantity, the preferred $\gamma$ changes across training rather than staying fixed. The figure is here: https://ibb.co.com/bjFTrcCR
>
> We agree that this does **not** amount to estimating exact global constants for the 124M LLM run. Our claim is narrower: the new experiment provides a concrete **local/trajectory-level validation** of the mechanism in the theorem and shows that the theory-derived $\gamma^\star$ can be informative in practice.
>
> > The analysis covers vanilla MVR, not MARS-AdamW. Does it extend to MARS-AdamW / preconditioned space?
>
> This is an important scope point. Our LLM experiment indeed uses **MARS-AdamW**, but the theorem intentionally analyzes **vanilla two-gradient $\gamma$-MVR**. This was a deliberate choice to isolate the effect of scaling the momentum correction term itself and to explain, in the cleanest possible setting, why $\gamma<1$ can be provably advantageous over $\gamma=1$. We will make this scope distinction clearer.
>
> We prefer not to overclaim here: extending $\gamma$-similarity to adaptive/preconditioned geometry is natural and important, but is **outside the present theorem claims**. In other words, the submission’s formal claim is not “we fully analyze MARS-AdamW”; it is “we identify the previously missing mechanism that can make scaled momentum correction provably better than MVR.”
>
> > The theory seems specific to MARS and does not obviously generalize.
>
> The key transferable insight is methodological: for scaled correction methods, the right object is not “standard similarity + an extra penalty,” but the aggregate quantity
> $\gamma d_\xi-d$.
> This is exactly what standard similarity fails to capture for MARS. We agree that formal extensions to other algorithms are future work, but we believe this viewpoint is broader than this single optimizer.
>
> > What properties make $\delta_\gamma<\delta$?
>
> Our answer is not limited to quadratics. Lemma 1 shows that, for any fixed pair $(x,y)$,
> $h(x,y;\gamma)=\gamma^2\mathbb{E}\|d_\xi-d\|^2+(1-\gamma)^2\|d\|^2$,
> so $\gamma$ trades off stochastic gradient-difference dispersion against the deterministic full-gradient difference. Lemma 2 then turns this into the computable upper bound
> $\delta_\gamma^2 \le \gamma^2\delta^2+(1-\gamma)^2L^2$.
> Example 1 provides a tight exact quadratic instance, but the underlying mechanism is not restricted to quadratics.
>
> Finally, regarding the early/late ordering, yes: the theory predicts **stage dependence**, not a universal monotone rule. Different training stages can favor different $\gamma$ because the local balance between the $\delta_\gamma$-driven gain and the $(1-\gamma)$ penalty changes along the trajectory, which is exactly what the new auxiliary measurements illustrate.

---

> > ### Author Rebuttal · Reviewer_EpCR · 2026-04-06
> >
> > I thank the authors for the response. The auxiliary CNN/CIFAR-10 experiment is a useful addition, but I note that the claim of $\gamma^* \sim 0.5$ appears to reflect only the early training phase — the figure suggests the estimated $\gamma^*$ drops substantially as training progresses, which is not fully consistent with the stated summary. . I would suggest the authors discuss this nuance more carefully when incorporating the result into the revision. That said, the core theoretical contribution is genuine and I am raising my score from 3 to 4.

---

### Official Review · Reviewer_wLzh · 2026-03-16

**Soundness:** 4
**Presentation:** 4
**Significance:** 3
**Originality:** 3
**Overall Recommendation:** 5
**Confidence:** 2

**Summary:**

This is a theory paper. MARS is an incremental gradient method that uses momentum to reduce the variance from mini-batch noise.
The paper aims to provide a better understanding of MARS's convergence. The main contribution lies in a new similarity condition: \gamma-similiarity, where \gamma is a MARS specific hyperparameter that adjusts the weight of the momentum correction term.

Prior works' analysis have two major limitations:
- Assuming inaccessible and time-varying values of \gamma for the convergence proof,
- The practical implication trick of re-using old gradients can hurt the convergence rate.

The paper addresses these two limitations, and then shows empirically that an optimal gamma enables MARS to show better performance compared with AdamW and MARS with \gamma = 1.

**Compliance With Llm Reviewing Policy:**

Affirmed.

**Final Justification:**

I maintain my positive impression of the paper after rebuttal. I would be glad to see the authors take into account the rebuttal contents in the revision.

**Key Questions For Authors:**

1. In the LLM pre-training experiments, are any Adam style adaptive-step-size / normalization used for MARS? It is shocking to me that without any normaliztion, MARS can train LLM without stability issue.

2. I am not very familiar with the theory side of optimizaiton works but I have worked on control variate, and from my understanding \gamma has a purpose similiar to the coefficient in front of the control variate: If the correction term correlates with randomness in the gradient, then the coefficient should be large, and if not, it should be down weighted, otherwise it would slow down the convergence. Therefore I think optimal \gamma should be time varying as effectiveness of the correction term (\Delta(x_t, x_{t-1} ) ) can be different across training stages. I wonder what authors think about this?

**Limitations:**

Limitations are not discussed in the paper.

From my perspective, the limitation lies in the gap between the theory and practical applications: I didn't see how the derived theory converts anything practitioners can use, e.g. guideline or heuristics for setting \gamma.

**Strengths And Weaknesses:**

I am not an expert in optimization theory, so I am not sure how constructive my comments are. I also did not go through the theory parts in detail for correctness check.

# Strengths
- MARS is a promising direction: Incremental gradient methods are incompatible with data augmentation and didn't seem to show much value in old deep learning workloads such as ResNet or ViT. However the situation seems to different for LLM pre-training, methods such as MARS start to demonstrate again the value of controlling mini-batch noise. Therefore I believe the problem studied by the submission has deep connection with frontier training challenges.

- The paper is well motivated: I find the authors' claims on the limitations of prior theoretical works reasonable.

# Weakness
- I understand that this is a theory focused paper and empirical results may not be the focus. However, the results in Figure.2 seems to just be a replication of MARS and I didn't see clear relationship between the experiments and the paper's theoretical contribution.

---

> ### Author Rebuttal · Authors · 2026-03-31
>
> Thank you for the thoughtful review and for recognizing both the motivation and the practical relevance of understanding MARS in modern LLM pretraining.
>
> > In the LLM pre-training experiments, are any Adam-style adaptive-step-size / normalization used for MARS?
>
> Yes. Our LLM experiment uses MARS-AdamW, not vanilla MARS. More precisely, we adopt the public pretraining setup of Semenov et al. (2025) and compare MARS-AdamW against MVR-AdamW and AdamW under the same Adam-style preconditioned optimizer stack and the same fixed non-$\gamma$ hyperparameters. Our goal in Figure 2 was to isolate the effect of the MARS scaling coefficient $\gamma$ inside a realistic, already-tuned LLM training pipeline, rather than to claim that unnormalized MARS alone can train LLMs stably.
>
> > I think the optimal $\gamma$ should be time-varying.
>
> We agree that this is a very interesting direction. One qualitative implication of our analysis is precisely that the preferred $\gamma$ need **not** be universal across training. Proposition 1 shows that the benefit of $\gamma<1$ depends on the regime through the trade-off between the $\delta_\gamma$-driven gain and the $(1-\gamma)$-dependent penalty. This is consistent with the empirical observation in Figure 2 that the early-stage and late-stage orderings of $\gamma$ can differ.
>
> At the same time, our present goal is narrower and, in our view, foundational: to show for the first time that a **fixed practical $\gamma<1$ can be provably better than $\gamma=1$**, whereas prior theory either gave no separation between MARS and MVR or required infeasible time-varying coefficients depending on full-gradient quantities.
>
> To partially address the practical question, we have now run an additional auxiliary experiment on a small CNN on CIFAR-10 setup in the exact theorem-aligned setting. There we estimate checkpoint-level smoothness and similarity statistics from gradient differences, derive a theory-predicted $\gamma^\star$, and compare against a small $\gamma$ grid. Preliminary results show:
> (i) the measured preferred $\gamma$ is **not constant** over training;
> (ii) the theory-predicted run-level $\gamma^\star$ lies around **$0.5-0.6$**; and
> (iii) values in this range are close to the empirically best fixed $\gamma$ in that setup.
> We include the plot here: https://ibb.co.com/bjFTrcCR
>
> > Limitations are not discussed.
>
> We agree and will make this explicit. In the revision/camera-ready, we will add a dedicated limitations paragraph/section.
>
> More broadly, we agree with your main concern that theory should eventually produce practitioner-facing guidance. Our central claim here is not yet a full practical recipe, but a theoretical one that was missing before: **why and when $\gamma<1$ can provably help at all**. We hope the new checkpoint-level empirical measurements make that bridge more visible.

---

> > ### Author Rebuttal · Reviewer_wLzh · 2026-04-02
> >
> > I would like to thank the authors for the response!
> >
> > My questions are fully addressed, I will maintain my original score.

---

### Decision · Program_Chairs · 2026-04-30

**Decision:**

Accept (regular)

**Comment:**

This paper presents a theoretical analysis of MARS, a variance-reduction method for training LLMs. The main contribution is the introduction of a loss landscape assumption, termed $\gamma$-similarity, to characterize the interaction between the gradient noise structure and a key scaling factor in MARS. Based on this condition, the paper establishes a convergence guarantee for MARS.

Overall, the paper takes a meaningful step towards bridging the gap between the theory and empirical behavior of MARS. Reviewers were generally positive about the paper, but they also raised concerns about the lack of empirical validation for the $\gamma$-similarity assumption. During the rebuttal, the authors provided additional experiments on CIFAR-10 CNNs to support this assumption. Although these experiments are limited in scale, the reviewers generally found them sufficient to address the main concern.

Several minor weaknesses remain, including the mismatch between the use of SGD in the theoretical analysis and Adam in the experiments, as well as the lack of direct practical implications. However, these limitations are acceptable in the context of a theory paper and do not outweigh its contributions.

Therefore, we recommend acceptance of this paper.